# Laminar-specific cortico-cortical loops in mouse visual cortex

**Hedi Young, Beatriz Belbut, Margarida Baeta, Leopoldo Petreanu\***

Champalimaud Research, Champalimaud Center for the Unknown, Lisbon, Portugal

**Abstract** Many theories propose recurrent interactions across the cortical hierarchy, but it is unclear if cortical circuits are selectively wired to implement looped computations. Using subcellular channelrhodopsin-2-assisted circuit mapping in mouse visual cortex, we compared feedforward (FF) or feedback (FB) cortico-cortical (CC) synaptic input to cells projecting back to the input source (looped neurons) with cells projecting to a different cortical or subcortical area. FF and FB afferents showed similar cell-type selectivity, making stronger connections with looped neurons than with other projection types in layer (L)5 and L6, but not in L2/3, resulting in selective modulation of activity in looped neurons. In most cases, stronger connections in looped L5 neurons were located on their apical tufts, but not on their perisomatic dendrites. Our results reveal that CC connections are selectively wired to form monosynaptic excitatory loops and support a differential role of supragranular and infragranular neurons in hierarchical recurrent computations.

## Introduction

The complex network of cortical areas can be hierarchically ordered based on the anatomy of inter-areal cortico-cortical (CC) projections. Lower areas send bottom-up 'feedforward' (FF) inputs to higher areas, which reciprocate with anatomically distinct top-down 'feedback' (FB) inputs, while areas at the same hierarchical level interact through 'lateral' or 'mixed' inputs (*Felleman and Van Essen, 1991*; *Markov et al., 2013*). The fact that a similar hierarchical architecture is observed across areas and across species (*D'Souza et al., 2020*; *Felleman and Van Essen, 1991*; *Harris et al., 2019*) suggests that FF and FB interactions reside at the core of cortical function. As with local cortical circuits, posited to implement a conserved canonical computation in different cortical areas (*Douglas and Martin, 2004*; *Harris and Shepherd, 2015*), long-range cortical connections could be performing stereotyped functions in different areas and in different species. However, the precise function of FF and FB interactions in hierarchical processing remains poorly understood.

Several theories of hierarchical computation involve looped interactions between areas, in which FF and FB pathways selectively influence each other in a bidirectional manner (*Bastos et al., 2012*; *Guerguiev et al., 2017*; *Keller and Mrsic-Flogel, 2018*; *Lillicrap et al., 2016*; *Rao and Ballard, 1999*; *Roelfsema and Holtmaat, 2018*; *Sacramento et al., 2018*). It is well established that cortical areas are densely interconnected and that FF pathways are always reciprocated by FB projections (*Felleman and Van Essen, 1991*; *Gămănuţ et al., 2018*; *Oh et al., 2014*; *Zingg et al., 2014*). However, whether CC connections are wired to selectively facilitate looped computations remains unknown.

One possibility is that CC inputs specifically modulate neurons projecting back to the source of those inputs (looped neurons) indirectly via intermediary inhibitory or excitatory cells in the local circuit. Another possibility, not mutually exclusive to the previous one, is that CC projections selectively synapse onto looped neurons directly to form interareal monosynaptic loops, which would be excitatory since most long-range cortical afferents are glutamatergic.

Cortical projection neurons can be divided into three broad classes: intratelencephalic (IT) neurons, which project to cortical areas, pyramidal tract (PT) neurons, which project to multiple

**\*For correspondence:**
leopoldo.petreanu@neuro.
fchampalimaud.org

**Competing interests:** The authors declare that no competing interests exist.

subcerebral areas including the midbrain, and corticothalamic (CT) neurons, which project predominantly to the thalamus (*Gerfen et al., 2018*; *Harris and Shepherd, 2015*). Thus, long-range CC projections could selectively participate in excitatory monosynaptic loops by preferentially contacting looped IT neurons, while avoiding neighboring non-looped IT, PT, and CT neurons. Previous studies have found that CC inputs form monosynaptic loops in sensorimotor (*Kinnischtzke et al., 2016*; *Mao et al., 2011*; *Yamawaki et al., 2016*), frontal (*Zhang et al., 2016*), and visual cortices (*Johnson and Burkhalter, 1997*). There is also evidence that some CC projections selectively innervate IT neurons over PT and CT neurons (*Kim et al., 2015*; *Kinnischtzke et al., 2016*; *Zhang et al., 2016*), but also evidence that some do not (*Yamawaki et al., 2016*). However, it remains unknown whether CC inputs selectively contact looped neurons over closely intermingled IT neurons projecting elsewhere (non-looped IT neurons), or whether they innervate IT neurons equally regardless of their projection pattern. Furthermore, in order to implement selective recurrent interactions, both FF and FB connections could be required to specifically engage with looped IT neurons. Yet, whether the selectivity of CC input between two areas is similar for both ascending and descending projections or whether it varies also remains unknown.

Here, we measured the strength of CC afferents to different types of projection neurons in mouse visual cortex to test whether they are wired to specifically engage in monosynaptic looped interactions. Using a combination of subcellular channelrhodopsin-2 (ChR2)-assisted circuit mapping (sCRACM) (*Petreanu et al., 2009*) and injections of multiple retrograde tracers, we found that FF and FB axons selectively provide stronger inputs to looped neurons in layer (L) 5 and L6, while in L2/3, they remain either unselective for projection type or provide stronger inputs to non-looped neurons. Thus, both ascending and descending hierarchical streams display the same selectivity for specific looped projection neurons despite their different anatomical profiles. Moreover, preferential innervation of looped L5 neurons often involved synapses made on their apical, but not basal, dendrites.

## Results

### Neurons with different projection patterns are intermingled in visual areas

We studied CC connections between primary visual cortex (V1), the lowest-order area of mouse visual cortex, and either the lateral visual areas (V2L) or the medial visual areas (V2M). Using dual injections of retrograde tracers, we measured the laminar distribution of different projection neurons in V1 and V2L (*Figure 1*, *Figure 1—figure supplement 1*). In each experiment, we compared the laminar distribution of V2L- or V1-projecting IT neurons with either IT neurons projecting to V2M, PT neurons projecting to the superior colliculus (SC), or CT neurons projecting to the visual thalamus. In each case, the different projection neurons were closely intermingled (*Figure 1A,B*). In both V1 and V2L, IT neurons were distributed across all layers except L1, including L4 (*Harris et al., 2019*; *Minamisawa et al., 2018*), indicating that FF and FB projections originate from neurons spanning most of the cortical depth. In contrast, PT and CT neurons were confined to L5 and L5/6, respectively, as previously described (*Harris and Shepherd, 2015*; *Figure 1A,B*). In both V1 and V2L, we found double-labeled IT neurons in L2–6 after injecting tracers in two different cortical areas, indicating that subpopulations of ascending and descending projection neurons have diverging axons innervating more than one visual cortical area (*Han et al., 2018*). However, IT neurons were rarely double-labeled when tracers were injected in a cortical and subcortical area, confirming that cortical- and subcortical-projecting neurons constitute different classes of projection neurons (*Economo et al., 2018*; *Harris and Shepherd, 2015*; *Tasic et al., 2018*). Using AAV encoding green fluorescent protein (AAV-GFP), we anterogradely traced V1→V2L FF axons and V2L→V1 FB axons and measured their laminated termination pattern (*Figure 1C,D*). FF axons in V2L were present in all layers but were denser in L2/3 and L6. FB axons in V1 arborized in L1 and L6, while avoiding middle layers. Given the laminar distribution of the different projection neurons and the termination pattern of CC afferent axons, FF and FB projections could potentially directly innervate both looped and non-looped IT neurons in each cortical layer, as well as PT and CT neurons in L5 and L6 (*Figure 2A*). Moreover, the proximity of looped IT neurons and other projection classes indicates that FF and FB axons are equally accessible to them. Thus, functional mapping of FF and FB connections is required

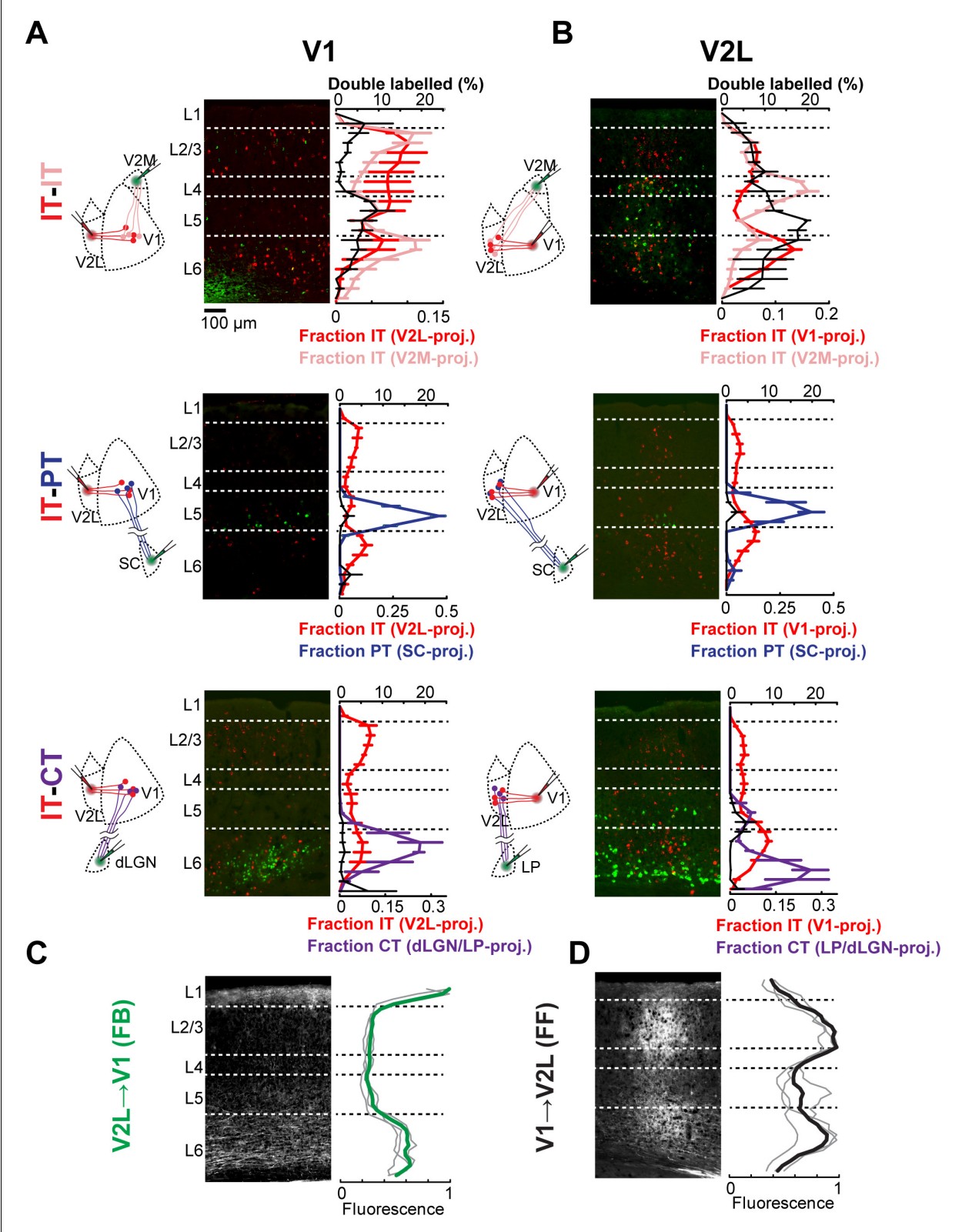

**Figure 1.** Cortical neurons projecting to different areas are intermingled and accessible to feedforward (FF) and feedback (FB) axons. (**A**) Distribution of retrogradely labeled projection neurons in primary visual cortex (V1) after injection of a red-fluorescent tracer in lateral visual areas (V2L) and an infrared-fluorescent tracer in either medial visual areas (V2M), superior colliculus (SC), or visual thalamus. Left, experimental configuration; center, representative fluorescent histological section, with infrared fluorescence shown in green; right, colored traces show the mean laminar distribution of

*Figure 1 continued on next page*

*Figure 1 continued*

the different projection neurons binned in 50 µm increments, while the black trace shows the percentage of retrogradely labeled neurons that are double-labeled at each depth (n = 3 animals per group). Error bars, standard error; dashed lines, approximate layer boundaries. (B) Distribution of retrogradely labeled projection neurons in V2L after injection of a red-fluorescent tracer in V1 and an infrared-fluorescent tracer in either V2M, SC, or visual thalamus (n = 3 animals per group). (C) Distribution of anterogradely labeled V2L FB axons in V1. Left, representative fluorescent histological section; right, axonal fluorescence across cortical depth binned in 50 µm increments. Individual mice, thin gray traces; average, thick green trace (n = 3 animals). (D) Distribution of anterogradely labeled V1 FF axons in V2L (n = 3 animals).

The online version of this article includes the following figure supplement(s) for figure 1:

**Figure supplement 1.** Histological and in vivo verification of lateral visual (V2L) and medial visual (V2M) area injection sites.

to reveal any underlying synaptic specificity, since axo-dendritic overlap does not always predict connectivity (*Harris and Shepherd, 2015*).

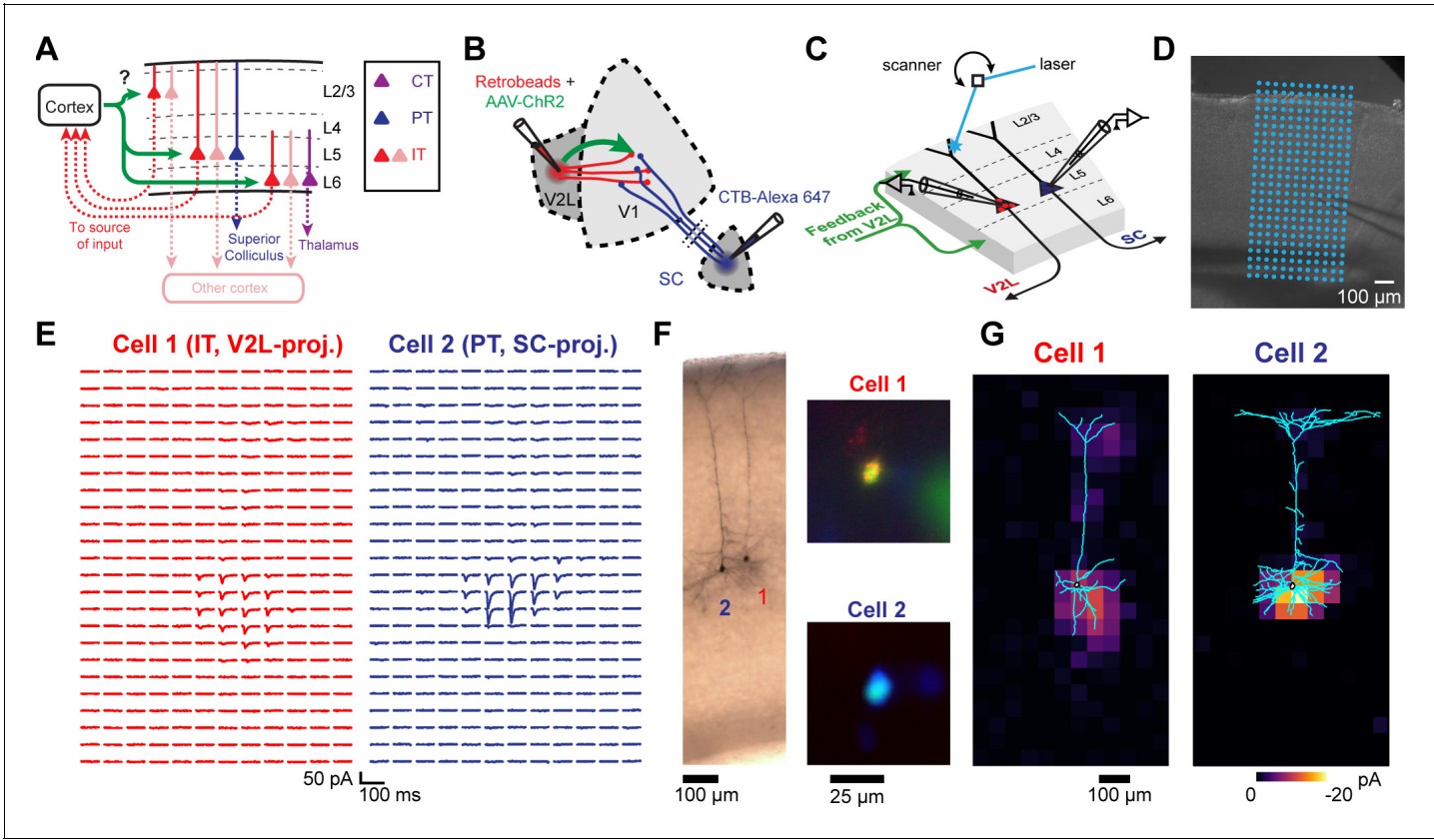

**Figure 2.** Measuring the strength and dendritic distribution of cortico-cortical (CC) inputs to different projection neurons. (A) We probed the strength of CC inputs to looped and non-looped neurons in different cortical layers. (B) Example experiment configuration. Retrograde tracers are injected in two areas to label different projection neurons. One cortical area is also co-injected with adeno-associated virus (AAV)-channelrhodopsin-2 (ChR2) to express ChR2 in a specific CC projection. (C) Example of subcellular channelrhodopsin-2 (ChR2)-assisted circuit mapping (sCRACM) experiment. Pairs of neighboring retrogradely labeled neurons in the same cortical layer were sequentially recorded. During each recording, a laser beam was scanned over the dendrites of the cell at different locations in a grid pattern. (D) Brightfield image of an acute coronal cortical slice showing the recording pipette and photostimulation grid. (E) Excitatory postsynaptic currents (EPSCs) recorded from a pair of neighboring L5 neurons, evoked by photostimulating ChR2⁺ V2L→V1 FB terminals on a grid. (F) Left, dendritic morphology staining of the recorded pair. Right, identity of the recorded projection neuron was confirmed by fluorescence in the soma of both a retrograde tracer and a different-colored dye introduced from the internal patch pipette solution. (G) sCRACM maps of the recorded pair overlaid on their reconstructed dendrites. Responsive locations are color-coded to represent mean amplitude.

The online version of this article includes the following figure supplement(s) for figure 2:

**Figure supplement 1.** Total input vs. laminar depth across different projections and projection neuron classes.
**Figure supplement 2.** Analysis of the incidence of retrograde infection of projection neurons by adeno-associated viruses (AAVs).

## Mapping the cell-type specificity of CC connections

To measure the cell-type selectivity of CC connections, we combined sCRACM (*Petreanu et al., 2009*) with multiple injections of fluorescent retrograde tracers (*Figure 2B*). We injected AAV-ChR2 mixed with a retrograde tracer in either V1, V2L, or V2M to express ChR2 in FF or FB axons and to retrogradely label looped IT neurons projecting back to the source of ChR2-expressing (ChR2$^+$) axons. We also injected a different retrograde tracer in either a second cortical area, the SC, or the thalamus to label non-looped IT, PT, or CT neurons (*Figure 2A*). We recorded from pairs of neighboring neurons in the same cortical layer in either V1 or higher-order visual areas V2L or V2M in acute brain slices containing FB or FF ChR2$^+$ axons, respectively. For each pair, one cell projected to the source of ChR2$^+$ inputs and one cell projected to a different cortical or subcortical area, as indicated by the retrograde tracer type in the soma (*Figure 2C,F*). Double-labeled cells were excluded. During the recording, we used galvanometer mirrors to rapidly photostimulate ChR2$^+$ terminals with a blue laser at different locations around the cell to evoke excitatory postsynaptic currents (EPSCs) in the presence of sodium channel blocker tetrodotoxin (TTX), potassium-channel blocker 4-aminopyridine (4-AP), and the NMDA-receptor blocker 3-((R)-2-carboxypiperazin-4-yl)-propyl-1-phosphonic acid (CPP) (*Figure 2C–E*). The onset of laser-evoked EPSCs was delayed relative to laser pulse onset in 96% (225/235) of the recorded neurons labeled with retrograde tracers, indicating that retrograde AAV-mediated transfection of ChR2 was rare, consistent with histological analyses (Materials and methods, *Figure 2—figure supplement 2*). Thus, sCRACM maps provide a measure of both the strength and location of monosynaptic CC inputs with minimal contamination from local collaterals. We then compared the strength of monosynaptic FF or FB inputs in defined dendritic compartments in pairs of neurons projecting to different areas (*Figure 2E–G*; *D'Souza et al., 2016*; *Morgenstern et al., 2016*; *Petreanu et al., 2009*; *Yang et al., 2013*). We measured input strength of 4 CC projections (FF: V1→V2L and V1→V2M; FB: V2M→V1 and V2L→V1) to looped IT neurons in three layers (L2/3, L5, and L6), and compared it to non-looped IT, PT, or CT neurons in their vicinity (*Figure 2—figure supplement 1*).

## FF and FB inputs innervate specific dendritic compartments of projection neurons

We detected monosynaptic FF and FB inputs in every cell type and analyzed their dendritic distribution (*Figure 3*). Individual sCRACM maps were normalized to their maximum response, aligned relative to pia (*Figure 3*) or soma position (*Figure 3—figure supplement 1*) and averaged. They thus represent the relative distribution of CC inputs within the dendritic tree for each class of projection neuron. Input strength to distal dendrites is underestimated in sCRACM maps as inputs are filtered and attenuated when measured from the soma due to the passive cable properties of dendritic arbors (*Petreanu et al., 2009*; *Stuart and Spruston, 2015*; *Williams and Mitchell, 2008*). In L2/3, FF inputs were largely confined to perisomatic dendrites. In L6, while FF connections also targeted perisomatic dendrites in both CT and IT neurons, we could detect additional FF input on the apical dendrites of IT neurons extending across L5 (*Figure 3A*, *Figure 4F*). Similarly, FF input contacted L5 IT neurons in both the perisomatic dendrites and along their apical dendritic trunk spanning L4 and L2/3 (*Figure 3A*, *Figure 5B*). Thus, like other long-range inputs (*Petreanu et al., 2009*), FF axons innervate several dendritic domains of L5 pyramidal neurons, revealing that targeting of apical dendrites is not an exclusive property of FB axons (see below).

As with FF inputs, FB afferents innervated the perisomatic compartments of all recorded cell types. However, when compared to FF inputs, a larger fraction of FB to L2/3 and L5 neurons was located on their distal dendrites in L1 (*Figure 3B*), consistent with previous measurements (*Petreanu et al., 2009*; *Yang et al., 2013*) and with the laminar profile of FB axons (*Figure 1C*). Apical tuft inputs were more readily detected in L5 IT neurons than in PT neurons, suggesting differential innervation by FB fibers (*Figure 2G*, *Figure 3B* and *Figure 5N*, see below).

## CC inputs are selectively stronger in looped L6 neurons

We first measured the connectivity of FF and FB inputs to L6 neurons projecting to different areas. We compared the input strength to looped IT neurons (i.e., neurons projecting back to the FF or FB input source) vs. either neighboring non-looped IT neurons projecting to another cortical area or CT neurons (*Figure 4*). We measured both total input strength, summing responses over all

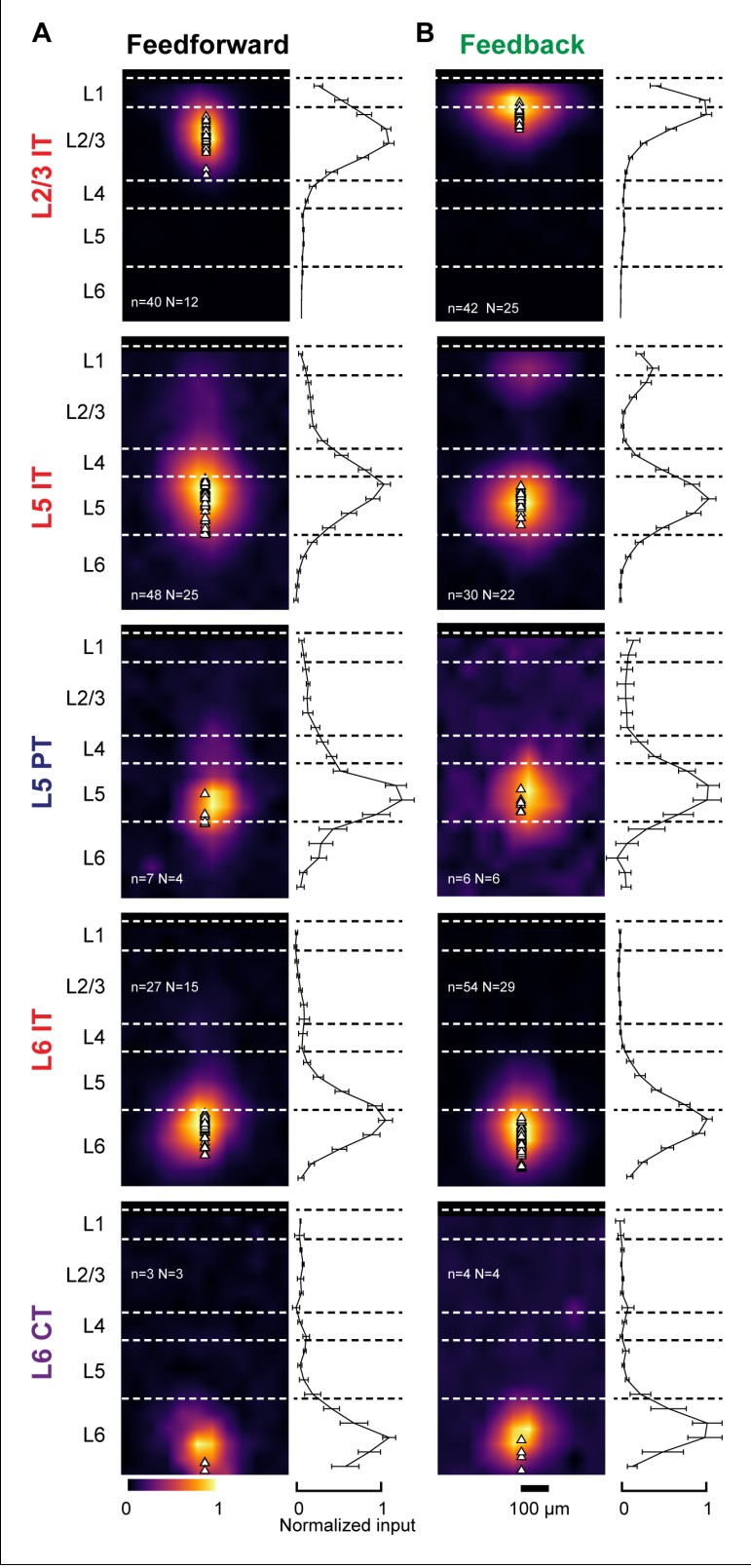

**Figure 3.** Dendritic distribution of feedforward (FF) and feedback (FB) inputs to different projection neuron classes. (**A**) Left, group averages of subcellular channelrhodopsin-2 (ChR2)-assisted circuit mapping (sCRACM) maps aligned by pia position showing primary visual cortex (V1) FF input to the different cell types (combining V1→V2L and V1→V2M inputs in the case of intratelencephalic [IT] neurons). Triangles, soma position. Right, *Figure 3 continued on next page*

*Figure 3 continued*

vertical profiles of input strength. Error bars, s.e.m.; n, number of neurons; N, number of mice. (**B**) Group averages and vertical profiles of sCRACM maps showing FB input to the different cell types in V1 (combining V2L→V1 and V2M→V1 inputs in the case of IT neurons).

The online version of this article includes the following figure supplement(s) for figure 3:

**Figure supplement 1.** Soma-aligned dendritic distribution of feedforward (FF) and feedback (FB) inputs to different projection neuron classes.

---

photostimulated locations, and input strength to the perisomatic area (L6 and lower L5) and apical dendritic area (upper L5 and L4). Relative input strength between the looped IT neuron and the paired neuron was quantified using the sCRACM Response Index (SRI), which ranges from −1 to 1. The SRI has a value of 0 if input strength in the two cells is equal, −1 if there is input in the looped IT neuron but no input in the paired neuron, and 1 in the opposite case (see Materials and methods).

We first compared V1 FF inputs to looped and non-looped L6 IT neurons. For V1→V2L inputs, total strength was greater in looped IT neurons (*Figure 4—figure supplement 1*; SRI: −0.49 ± 0.23, p=0.002). This stronger innervation was present in both perisomatic (SRI: −0.50 ± 0.21, p=0.0001) and apical areas (SRI: −0.45 ± 0.45, p=0.0364) (*Figure 4A–D*). For V1→V2M inputs, we did not detect a difference in total strength between looped and non-looped IT neurons (*Figure 4—figure supplement 1*, SRI: −0.32 ± 0.48, p=0.064), but looped IT neurons again received significantly stronger input in the perisomatic area (*Figure 4C*; SRI: −0.36 ± 0.46, p=0.038). V1→V2M inputs in apical dendrites were mostly weak and not different between cell types (Apical SRI: 0.33 ± 0.69, p=0.247). We next compared V1 FF inputs to looped L6 IT neurons and same-layer CT neurons in V2L. Total input to looped neurons was consistently stronger than to CT neurons (*Figure 4—figure supplement 1*, SRI: −0.64 ± 0.23, p=1.2×10$^{-5}$) and inputs were weaker in both perisomatic and apical areas of CT neurons (*Figure 4E–H*, Perisomatic SRI: −0.64 ± 0.26, p=2.4×10$^{-5}$; Apical SRI: −0.69 ± 0.40, p=0.018).

Likewise, FB inputs to V1 also selectively targeted looped L6 neurons in many cases (*Figure 4I–P*). For the looped vs. non-looped comparison among IT neurons, FB from V2M preferentially connected to IT neurons projecting back to the source of FB, both when measuring total and perisomatic input (*Figure 4—figure supplement 1*, *Figure 4I–L*, Total SRI: −0.39 ± 0.36, p=0.031; Perisomatic SRI: −0.40 ± 0.36, p=0.028), but no difference was detected in the apical area (SRI: −0.27 ± 0.85, p=0.56). FB from V2L was not significantly different in looped and non-looped L6 IT neurons (total SRI: −0.18 ± 0.35, p=0.098; perisomatic SRI: −0.14 ± 0.35, p=0.182; apical SRI: −0.02 ± 0.90, p=0.958). However, looped neurons received stronger perisomatic input from V2L compared to neighboring CT neurons (*Figure 4M–P*; total SRI: −0.40 ± 0.42, p=0.003, perisomatic SRI: −0.41 ± 0.45, p=0.005). Thus, we found that in most experiments (three out of four), FF and FB connections to L6 neurons were significantly stronger in looped IT cells than in non-looped ones. FF and FB connections were also stronger in looped IT neurons than in CT neurons.

## FF and FB inputs are selectively stronger in looped L5 neurons

We next asked whether the strength of CC inputs differed among L5 pyramidal neuron types (*Figure 5*). FF and FB inputs innervated L5 neurons in both perisomatic and apical dendritic regions (*Figure 3*, *Figure 3—figure supplement 1* and *Figure 5B,F,J,N*). As before, we analyzed input strength in these two innervation domains separately. First, we compared V1→V2L or V1→V2M FF inputs to looped IT neurons projecting back to V1 vs. non-looped IT neurons (*Figure 5A–D*). In V2L, total input strength did not differ between looped and non-looped L5 IT neurons (*Figure 5—figure supplement 1*; SRI: −0.03 ± 0.35, p=0.783), and neither did perisomatic input (*Figure 5C*; SRI: −0.02 ± 0.36, p=0.82). However, FF inputs terminating on the apical dendrites of L5 neurons were stronger in looped IT neurons (*Figure 5D*; SRI: −0.40 ± 0.43, p=0.009). In V2M, looped neurons received stronger V1 input than neighboring non-looped IT neurons on both their perisomatic and apical dendrites (*Figure 5—figure supplement 1* and *Figure 5C,D*; total SRI: −0.42 ± 0.32, p=0.002; perisomatic SRI: −0.42 ± 0.35, p=0.002; apical SRI: −0.37 ± 0.43, p=0.0174).

We then compared the strength of FF inputs to looped IT and PT neurons. FF fibers innervated the perisomatic dendrites of looped neurons more strongly, with no difference observed in apical

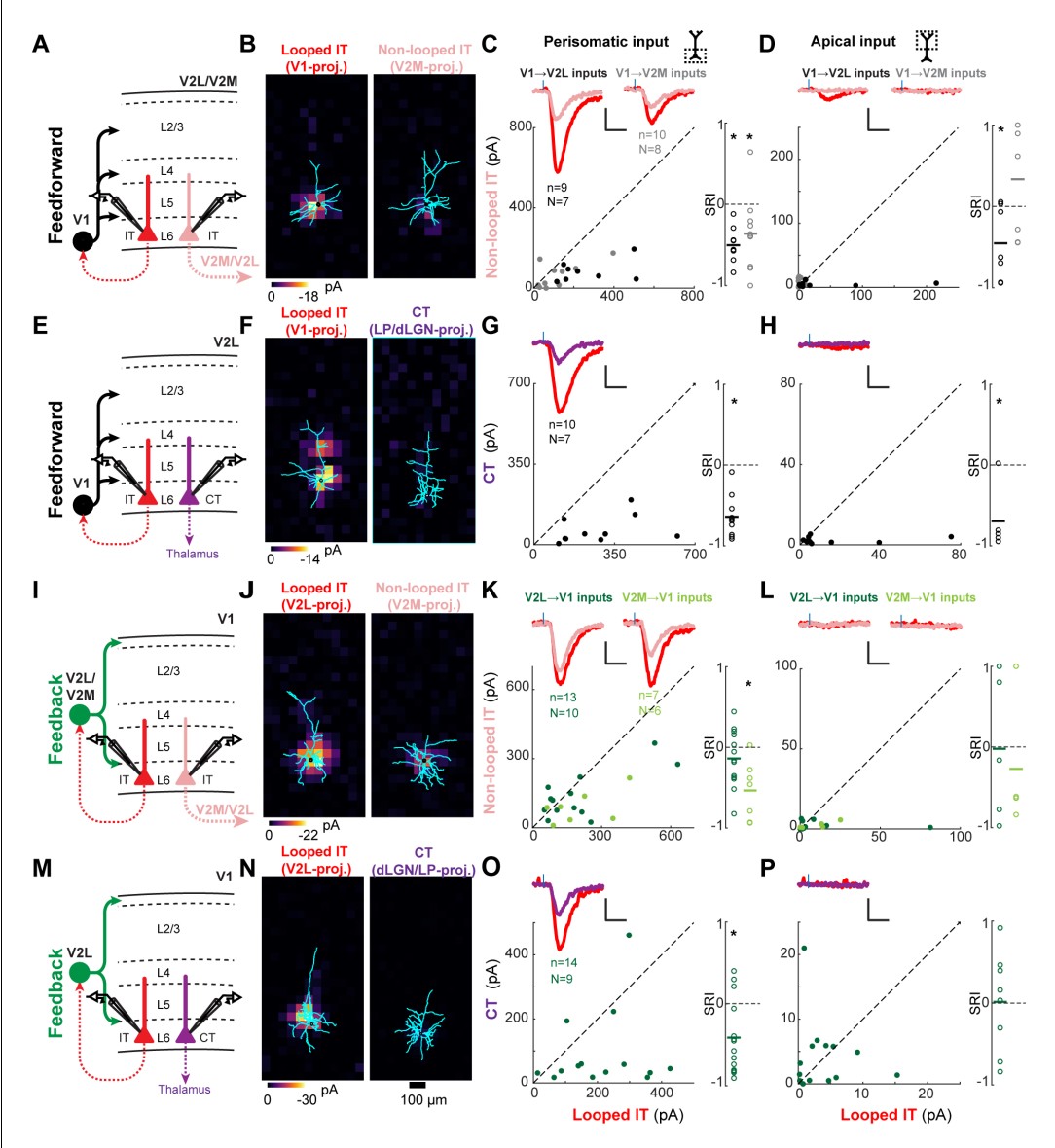

**Figure 4.** Most feedforward (FF) and feedback (FB) inputs are stronger in looped intratelencephalic (IT) neurons than in neighboring non-looped IT or corticothalamic (CT) neurons in L6. (**A**) Configuration of experiments comparing strength of primary visual cortex (V1) FF input to pairs of L6 looped and non-looped IT neurons in lateral visual area (V2L) or medial visual area (V2M). (**B**) Example pair of subcellular channelrhodopsin-2 (ChR2)-assisted circuit mapping (sCRACM) maps overlaid on reconstructed dendrites showing monosynaptic V1 FF inputs to a looped IT neuron (left) and an adjacent non-looped IT neuron (right) recorded in V2L. (**C**) Left, paired comparisons of perisomatic FF input to looped vs. non-looped IT neurons (n, number of cell pairs; N, number of mice); black dots, V1→V2L inputs; gray dots, V1→V2M inputs. Traces were generated by averaging the mean perisomatic excitatory postsynaptic current (EPSC) of each neuron across all neurons in the same projection class. Colors correspond to (**A**). Blue tick, laser pulse. Scale bars in all panels, 2 pA and 20 ms. Right, sCRACM Response Index (SRI) of the same data. Number of cell pairs and animals are the same as in the left plot unless otherwise specified. Horizontal line, mean. *, p<0.05, see text for exact value. (**D**) Same as C for apical inputs (SRI: V1→V2L, n = 7, N = 6; V1→V2M, n = 7, N = 6). (**E**) Configuration of experiment comparing strength of V1 FF input to pairs of L6 looped IT and CT neurons in V2L. (**F**) Example pair of sCRACM maps overlaid on reconstructed dendrites showing monosynaptic V1 FF inputs to a looped IT neuron (left) and an adjacent CT neuron (right) recorded in V2L. (**G**) Paired comparisons and SRI of perisomatic FF input to looped IT vs. CT neurons. (**H**) Paired comparisons and SRI (n = 5, N = 5) of apical FF input to looped IT vs. CT neurons. (**I**) Configuration of experiments comparing strength of V2L or V2M FB input to pairs of L6 looped and non-looped IT neurons in V1. (**J**) Example pair of sCRACM maps overlaid on reconstructed dendrites showing monosynaptic V2L FB inputs to a looped IT neuron (left) and an adjacent non-looped IT neuron (right) recorded in V1. (**K**) Paired comparisons and SRI of perisomatic FB input to looped vs. non-looped IT neurons. Dark green dots, V2L→V1 inputs; light green dots, V2M→V1 inputs. (**L**) Paired comparisons and SRI (V2L→V1, n = 5, N = 5; V2M→V1, n = 4, N = 4) of apical FB input to looped vs. non-looped IT neurons. (**M**) Configuration of experiment comparing strength of V2L FB input to pairs of L6 looped IT and CT neurons in V1. (**N**) Example pair of sCRACM maps overlaid on reconstructed dendrites showing monosynaptic V2L FB

*Figure 4 continued on next page*

*Figure 4 continued*

inputs to a looped IT neuron (left) and an adjacent CT neuron (right) recorded in V1. (**O**) Paired comparisons and SRI of perisomatic FB input to looped IT vs. CT neurons. (**P**) Paired comparisons and SRI (n = 8, N = 7) of apical FB input to looped IT vs. CT neurons.

The online version of this article includes the following figure supplement(s) for figure 4:

**Figure supplement 1.** Total subcellular channelrhodopsin-2 (ChR2)-assisted circuit mapping (sCRACM) input to L6 neurons.

inputs (*Figure 5—figure supplement 1* and *Figure 5E–H*; total SRI: −0.43 ± 0.41, p=0.003; periso-matic SRI: −0.46 ± 0.43, p=0.002; apical SRI: −0.14 ± 0.65, p=0.49). We conclude that FF inputs are selectively stronger in looped L5 neurons when compared to non-looped IT or PT neurons, and that this selectivity may result from inputs impinging on the perisomatic dendrites (V1→V2L, IT vs. PT), the apical dendrites (V1→V2L, IT vs. IT), or both regions (V1→V2M, IT vs. IT).

We next measured the strength of FB inputs to L5 neurons with different projection targets (*Figure 5I–P*). We transfected V2L or V2M FB axons with ChR2 and recorded different L5 cell types in V1. We first compared the strength of V2L→V1 and V2M→V1 FB inputs to looped vs. non-looped L5 IT neurons (*Figure 5I–L*). In both projections, total input (*Figure 5—figure supplement 1*; V2L→V1 SRI: −0.15 ± 0.54, p=0.284; V2M→V1 SRI: −0.19 ± 0.56, p=0.222) and perisomatic input (*Figure 5K*; V2L→V1 SRI: −0.09 ± 0.60, p=0.557; V2M→V1 SRI: −0.22 ± 0.57, p=0.152) were indis-tinguishable between the two projection types. However, inputs targeting the distal tufts in L1 were stronger in looped than non-looped IT neurons for both projections (*Figure 5L*; V2L→V1 SRI: −0.47 ± 0.51, p=0.012; V2M→V1 SRI: −0.48 ± 0.58, p=0.022). This resulted in looped neurons receiving a larger fraction of total FB input in their L1 apical domain (mean fraction of total input in L1; V2L→V1: looped, 23.1 ± 16.6%, non-looped, 11.0 ± 10.6%, p=0.01, signed-rank test; V2M→V1: looped, 18.1 ± 23.0%, non-looped, 9.4 ± 18.2%, p=0.01, signed-rank test). The presence of stronger apical inputs in looped neurons could not be explained by differences in dendritic filtering. First, V2L- and V2M-projecting populations had similar apical dendritic morphologies, with slender tufts that were indistinguishable in L1 (*Figure 5—figure supplement 2*). Second, the cell type receiving the largest fraction of distal input switched from V2L-projecting neurons in the V2L→V1 experiment to V2M-projecting neurons in the V2M→V1 experiment.

As with the looped vs. non-looped IT comparison, V2L→V1 total and perisomatic inputs did not distinguish between looped IT and PT neurons (*Figure 5—figure supplement 1*, *Figure 5M–P*; total SRI: −0.20 ± 0.52, p=0.195; perisomatic SRI: −0.19 ± 0.62, p=0.28), but looped neurons received stronger FB input in their L1 apical compartment (*Figure 5P*; SRI: −0.46 ± 0.62, p=0.026), despite having less total dendritic length in L1 (*Figure 5—figure supplement 2*). Simulations showed that the weaker sCRACM responses in distal tufts of PT neurons cannot be explained by differences in passive dendritic filtering between the two cell types. In the absence of connectional selectivity, the thicker apical shafts and richer apical tuft arborization of PT neurons (*Figure 5—figure supplement 2*) predict that distal FB input arriving at the soma would be larger, not smaller (*Figure 5—figure supplement 3*). We also confirmed that the weaker distal FB input in PT neurons was not due to dif-ferent levels of hyperpolarization-activated current ($I_h$) between the two cell types (*Harris and Shep-herd, 2015*), since looped IT neurons still received stronger FB input in the apical tuft when measured in the presence of $I_h$ blockers (*Figure 5—figure supplement 4*). Thus, the stronger sCRACM signals detected in L1 reflect synaptic selectivity for the terminal tufts of looped IT neurons. We conclude that the apical dendrites of L5 neurons have privileged access to FB axons when the neurons loop back to the source of those axons. Conversely, FB inputs to the basal dendrites of L5 neurons do not favor looped neurons over non-looped IT or PT neurons.

## The strength of CC inputs in looped and non-looped L2/3 neurons

Finally, we examined whether CC projections to supragranular neurons would also exhibit a prefer-ence for looped connectivity. In addition to total input strength (*Figure 6—figure supplement 1*), we analyzed inputs terminating in L1 and in the perisomatic region (*Figure 6*). FF input from V1 was equally strong in looped and non-looped L2/3 IT neurons in both V2L and V2M when assessing total input strength (*Figure 6—figure supplement 1*; V1→V2L SRI: −0.04 ± 0.30, p=0.65; V1→V2M SRI: −0.05 ± 0.35, p=0.683) and perisomatic innervation (*Figure 6C*; V1→V2L SRI:−0.06 ± 0.30, p=0.564; V1→V2M SRI: −0.17 ± 0.34, p=0.167). However, when considering FF connections in L1, V1→V2M

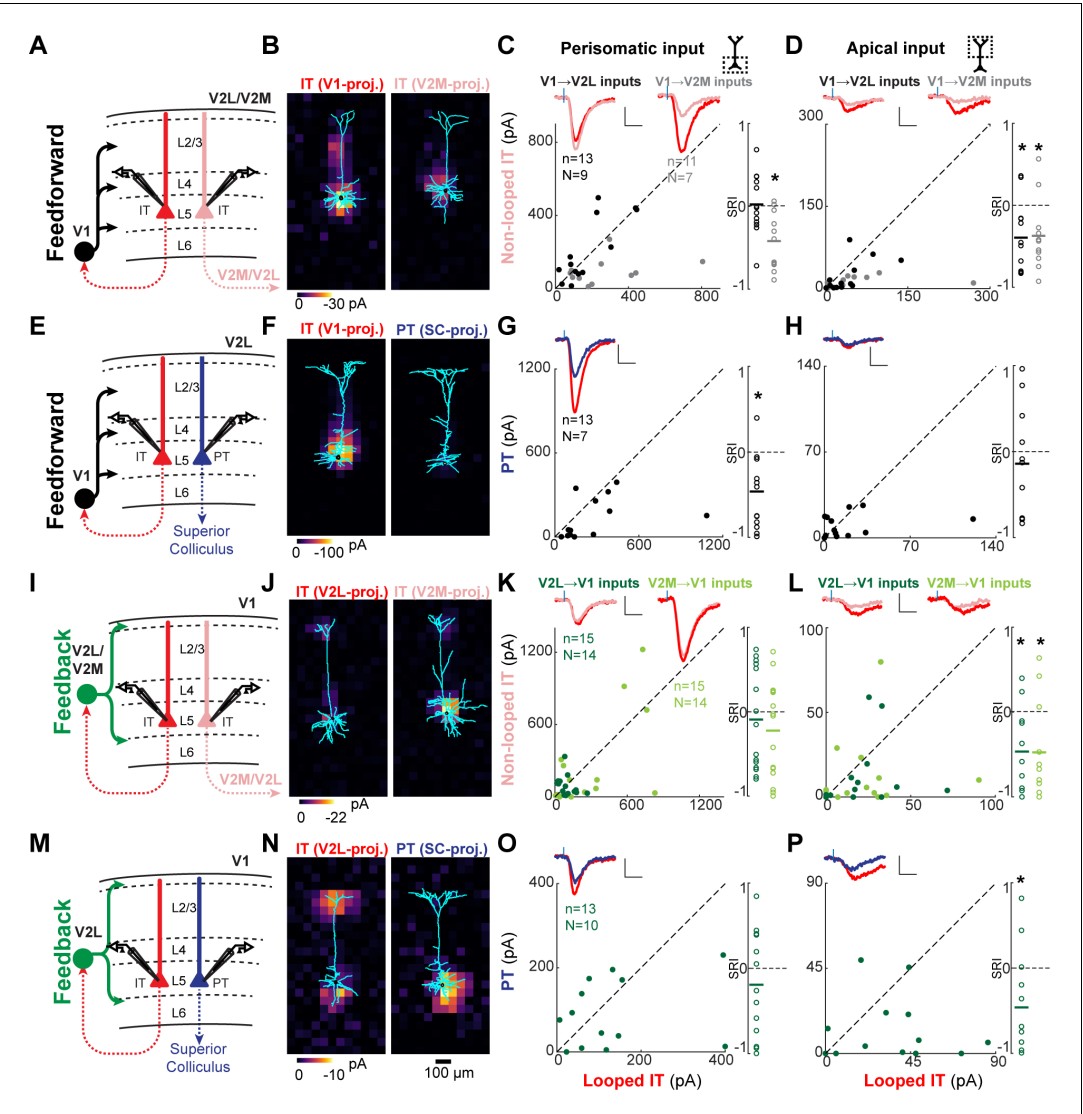

**Figure 5.** Feedforward (FF) and feedback (FB) inputs are stronger in looped intratelencephalic (IT) neurons than in neighboring non-looped IT or pyramidal tract (PT) neurons in L5. (A) Configuration of experiments comparing strength of primary visual cortex (V1) FF input to pairs of L5 looped and non-looped IT neurons in lateral visual (V2L) or medial visual (V2M) areas. (B) Example pair of subcellular channelrhodopsin-2 (ChR2)-assisted circuit mapping (sCRACM) maps overlaid on reconstructed dendrites showing monosynaptic V1 FF inputs to a looped IT neuron (left) and an adjacent non-looped IT neuron (right) recorded in V2L. (C) Left, paired comparisons of perisomatic FF input to looped vs. non-looped IT neurons; black dots, V1→V2L inputs; gray dots, V1→V2M inputs. Traces were generated by averaging the mean perisomatic excitatory postsynaptic current (EPSC) of each neuron across all neurons in the same projection class. Blue tick, laser pulse. Scale bars in all panels, 2 pA and 20 ms. Right, sCRACM Response Index (SRI) of the same data. Number of cell pairs and animals are the same as in the left plot unless otherwise specified. Horizontal line, mean. *, p<0.05, see text for exact value. (D) Same as C for apical inputs (SRI: V1→V2L, n = 12, N = 8; V1→V2M, n = 11, N = 7). (E) Configuration of experiment comparing strength of V1 FF input to pairs of L5 looped IT and PT neurons in V2L. (F) Example pair of sCRACM maps overlaid on reconstructed dendrites showing monosynaptic V1 FF inputs to a looped IT neuron (left) and an adjacent PT neuron (right) recorded in V2L. (G) Paired comparisons and SRI of perisomatic FF input to looped IT vs. PT neurons. (H) Paired comparisons and SRI (n = 11, N = 7) of apical FF input to looped IT vs. PT neurons. (I) Configuration of experiments comparing strength of V2L or V2M FB input to pairs of L5 looped and non-looped IT neurons in V1. (J) Example pair of sCRACM maps overlaid on reconstructed dendrites showing monosynaptic V2L FB inputs to a looped IT neuron (left) and an adjacent non-looped IT neuron (right) recorded in V1. (K) Paired comparisons and SRI of perisomatic FB input to looped vs. non-looped IT neurons. Dark green dots, V2L→V1 inputs; light green dots, V2M→V1 inputs. (L) Paired comparisons and SRI (V2L→V1, n = 11, N = 10; V2M→V1, n = 11, N = 10) of FB input in L1 to looped vs. non-looped IT neurons. (M) Configuration of experiment comparing strength of V2L FB input to pairs of L5 looped IT and PT neurons in V1. (N) Example pair of sCRACM maps overlaid on reconstructed dendrites showing monosynaptic V2L FB inputs to a looped IT neuron (left) and an adjacent PT neuron (right) recorded in V1. (O) Paired comparisons and SRI of perisomatic FB input to looped IT vs. PT neurons. (P) Paired comparisons and SRI (n = 12, N = 9) of FB input in L1 to looped IT vs. PT neurons.

*Figure 5 continued on next page*

*Figure 5 continued*

The online version of this article includes the following figure supplement(s) for figure 5:

**Figure supplement 1.** Total subcellular channelrhodopsin-2 (ChR2)-assisted circuit mapping (sCRACM) input to L5 neurons.

**Figure supplement 2.** Dendritic morphology of the different L5 projection neuron types in primary visual cortex (V1).

**Figure supplement 3.** Simulations of the dendritic filtering of distal apical inputs.

**Figure supplement 4.** Feedforward (FB) input to looped L5 intratelencephalic (IT) neurons vs. pyramidal tract (PT) neurons in the presence of $I_h$ blockers.

inputs were significantly stronger in non-looped neurons projecting to V2L, while V1→V2L inputs showed no preference (*Figures 6A,B,D*; V1→V2L SRI: −0.03 ± 0.66, p=0.892; V1→V2M SRI: 0.68 ± 0.52, p=0.013). This resulted in non-looped V2M L2/3 neurons receiving a larger fraction of total V1 FF input in L1 compared to their looped neighbors (mean fraction of total input in L1; V1→V2M inputs: looped, 2.1 ± 3.7%, non-looped, 12.6 ± 13.0%, p=0.039, signed-rank test). Similarly, we measured the strength of V2M and V2L FB inputs to different L2/3 IT neurons in V1. Total input was indistinguishable between looped and non-looped neurons, as were perisomatic and L1 inputs (*Figure 6—figure supplement 1*, *Figure 6E–H*; V2L→V1, total SRI: −0.13 ± 0.51, p=0.36, perisomatic SRI: −0.13 ± 0.54, p=0.394, L1 input SRI: −0.12 ± 0.64, p=0.553; V2M→V1, total SRI: −0.04 ± 0.36, p=0.67, perisomatic SRI: 0.03 ± 0.48, p=0.826, L1 input SRI: −0.04 ± 0.66, p=0.86). In summary, most (3/4) FF and FB inputs did not show projection-type specificity when innervating L2/3 neurons, with the exception of V1→V2M FF inputs, which were weaker, not stronger, when

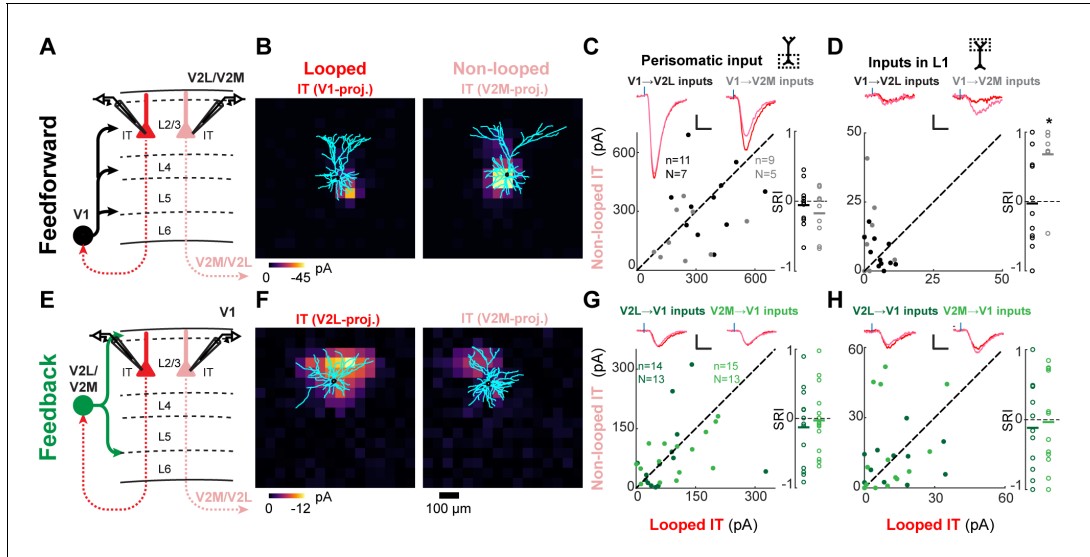

**Figure 6.** Feedforward (FF) and feedback (FB) connections are similar or weaker in looped L2/3 neurons. (**A**) Configuration of experiments comparing strength of primary visual cortex (V1) FF input to pairs of L2/3 looped and non-looped intratelencephalic (IT) neurons in lateral visual area (V2L) or medial visual area (V2M). (**B**) Example pair of subcellular channelrhodopsin-2 (ChR2)-assisted circuit mapping (sCRACM) maps overlaid on reconstructed dendrites showing monosynaptic V1 FF inputs to a looped IT neuron (left) and an adjacent non-looped IT neuron (right) recorded in V2L. (**C**) Left, paired comparisons of perisomatic FF input to looped vs. non-looped IT neurons; black dots, V1→V2L inputs; gray dots, V1→V2M inputs. Traces were generated by averaging the mean perisomatic excitatory postsynaptic current (EPSC) of each neuron across all neurons in the same projection class. Blue tick, laser pulse. Scale bars in all panels, 2 pA and 20 ms. Right, sCRACM Response Index (SRI) of the same data. Number of cell pairs and animals are the same as in the left plot unless otherwise specified. Horizontal line, mean. *, p<0.05, see text for exact value. (**D**) Same as C for inputs in L1 (SRI: V1→V2L, n = 11, N = 7; V1→V2M, n = 7, N = 5). (**E**) Configuration of experiments comparing strength of V2L or V2M FB input to pairs of L2/3 looped and non-looped IT neurons in V1. (**F**) Example pair of sCRACM maps overlaid on reconstructed dendrites showing monosynaptic V2L FB inputs to a looped IT neuron (left) and an adjacent non-looped IT neuron (right) recorded in V1. (**G**) Paired comparisons and SRI of perisomatic FB input to looped vs. non-looped IT neurons. Dark green dots, V2L→V1 inputs; light green dots, V2M→V1 inputs. (**H**) Same as G for inputs in L1 (SRI: V2L→V1, n = 11, N = 10; V2M→V1, n = 12, N = 11).

The online version of this article includes the following figure supplement(s) for figure 6:

**Figure supplement 1.** Total subcellular channelrhodopsin-2 (ChR2)-assisted circuit mapping (sCRACM) input to L2/3 neurons.

contacting dendrites of looped neurons in L1, in contrast to the looped preference observed in L5 and L6.

## Selectivity for looped IT neurons differs across infragranular and supragranular layers

We then searched for common patterns in the synaptic selectivity of CC projections to different IT neurons. In each layer, we plotted the strength of FF or FB projections to looped IT neurons relative to non-looped IT neurons in the perisomatic or apical compartments, as measured using the SRI (*Figure 7*). While there was some variability across individual projections as previously described, there were some common patterns shared across projections. When comparing inputs to looped and non-looped IT neurons in L5 and L6, in seven out of eight FF/FB projections we found stronger inputs to looped IT neurons, either in the perisomatic or apical areas, or both. On the contrary, none of the four FF/FB projections to L2/3 showed stronger inputs to looped IT neurons, and in the only case in which input strength differed between looped and non-looped neurons, inputs were weaker in looped neurons rather than in non-looped ones. When comparing looped IT neurons and PT or CT neurons, we found that in all cases CC inputs were stronger in looped IT neurons. We conclude that most FF and FB connections are selectively wired to strongly innervate looped IT neurons in L5 and L6. In L5, this selectivity involved synapses on apical, but not basal, dendrites in most cases (1/3 and

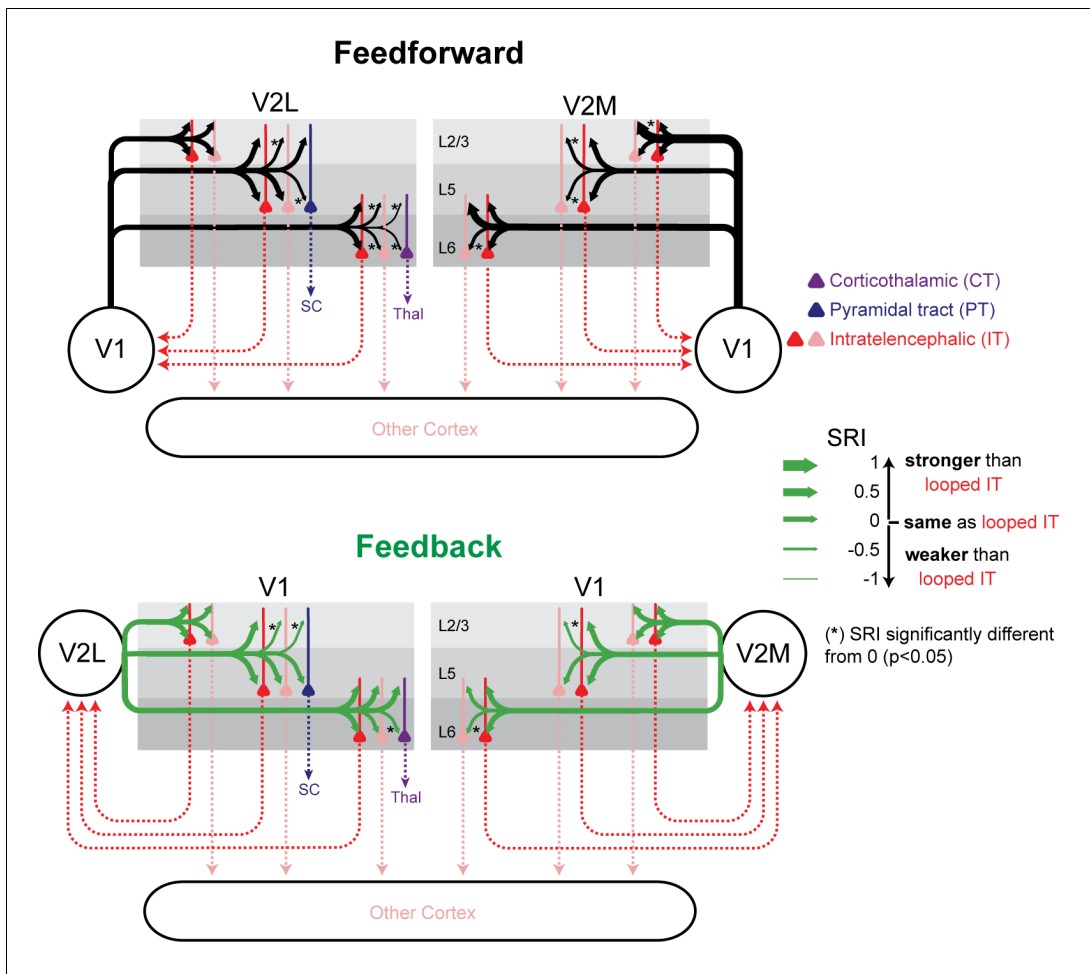

**Figure 7.** Summary of relative input strength across all experiments. The sCRACM Response Index (SRI) of feedforward (FF) and feedback (FB) inputs to the different cell types is represented by arrow thickness. The top and bottom arrows to L5 neurons indicate inputs to apical and perisomatic domains, respectively. Inputs to looped intratelencephalic (IT) cells in each cortical layer are assigned an SRI of 0 (medium arrow thickness). * signifies significant difference (p<0.05) from the looped IT population.

3/3 of FF and FB connections tested, respectively). By contrast, inputs to L2/3 were either equally strong in looped and non-looped IT neurons or weaker in looped ones.

## FB inputs selectively influence the activity of looped neurons in V1

These sCRACM measurements of input strength suggest that CC inputs exert different influences on their target neurons depending on their looped connectivity. To test the functional implications of the selective innervation of infragranular looped IT neurons, we performed experiments in current clamp in the absence of channel blockers. We injected the same AAV-ChR2 in V2M and measured voltage changes and spiking activity of V2M-projecting L6 neurons in V1 upon photostimulation, comparing their responses to those of neighboring V2L-projecting neurons (*Figure 8*). The resting potential of both projection types was similar (looped: −72.35 ± 5.85 mV, non-looped: −74.52 ± 3.58 mV, p=0.38). In the absence of current injection, a brief LED light pulse resulted in depolarizations, but not spiking activity, in the recorded cells under these illumination conditions. Depolarizations were larger in looped IT L6 neurons than in non-looped ones (*Figure 8A–C*; mean amplitude; looped IT, 10.2 ± 4.8 mV; non-looped IT, 4.3 ± 3.2 mV, p=0.004, signed-rank test). We compared the relative magnitude of the evoked postsynaptic potentials using the CRACM Response Index (CRI), analogous to the SRI used in the sCRACM experiments (Materials and methods). The relative difference in input strength between looped and non-looped cells measured with current-clamp LED-induced responses was similar to that measured with

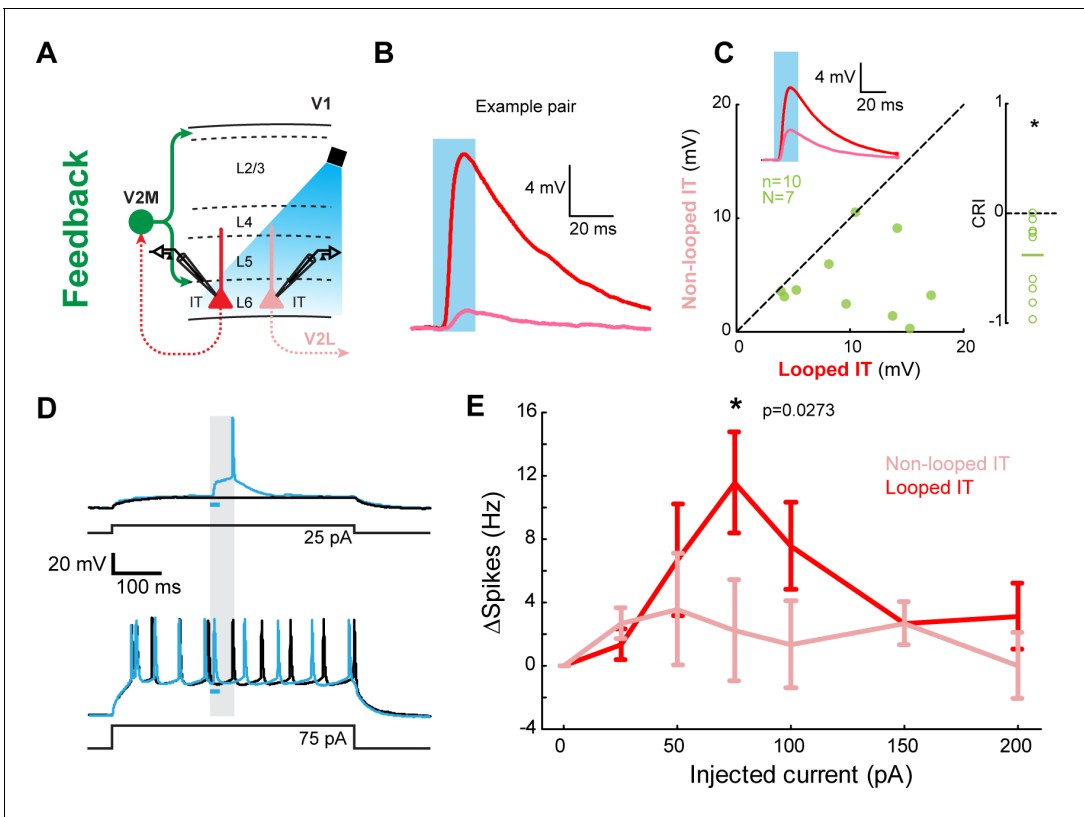

**Figure 8.** Feedback (FB) inputs in L6 can selectively modulate activity in looped intratelencephalic (IT) neurons. (**A**) Experiment configuration. In the absence of channel blockers, V2M→V1 FB axons were photostimulated using an LED during current-clamp recordings of looped and non-looped IT neurons in L6. (**B**) Example of excitatory postsynaptic potentials (EPSPs) from an example pair. Blue shade, light pulse. (**C**) Left, paired comparisons of EPSP amplitudes evoked in looped vs. non-looped IT neurons. Inset traces represent group averages for each projection class. Blue shade, light pulse. Right, CRACM Response Index (CRI) of the same data (n = 10, N = 7). (**D**) Example traces of FB modulation in a looped IT neuron. Cells were driven by a sustained positive current injection. Black traces, LED OFF trials; blue traces, LED ON trials. Blue bar, duration of the LED light pulse. Gray shading, time period used to analyze spiking activity in (**E**). (**E**) Spike rate difference between LED-ON and LED-OFF trials in looped and non-looped IT neurons as a function of the amount of current injected during the depolarization step. *, p=0.0273, paired t-test with Bonferroni correction for seven comparisons.

sCRACM (*Figure 8C*, *Figure 4—figure supplement 1* and *Figure 2—figure supplement 2*; current-clamp CRI: −0.38 ± 0.35; sCRACM total input SRI with AAV2/1: −0.39 ± 0.36; sCRACM total input SRI with AAV5: −0.52 ± 0.39), indicating that sCRACM reflects actual input strength in the functioning circuit. We then asked how V2M→V1 FB combines with bottom-up input in looped and non-looped L6 IT neurons. We injected steps of positive current at the soma through the recording pipette to mimic bottom-up depolarization, and measured evoked spiking activity while photostimulating FB axons (*Figure 8D–E*). Photoactivation of V2M→V1 axons induced additional spiking activity over a range of current injections (*Figure 8D–E*). Looped neurons spiked more frequently upon FB photoactivation than their non-looped neighbors (*Figure 8E*). Thus, stronger FB innervation of looped IT neurons in L6 can selectively modify their spiking activity when they are concurrently depolarized by other inputs.

## Discussion

We comprehensively measured the connection strength of several ascending and descending projections to the three major classes of cortical projection neurons across layers and across areas. We found that both FF and FB preferentially innervated looped over non-looped neurons in the infragranular layers, but not in the supragranular layers, resulting in a selective modulation of spiking activity in looped neurons. Furthermore, by mapping the dendritic locations of CC synaptic inputs, we show that targeting of looped neurons is often highly subcellular, with many projections showing selectivity for the apical domains of looped neurons, but not for their basal domains.

### Looped connectivity in CC interactions

Consistent with previous reports, we found that CC axons provide inputs to all neurons regardless of their projection type (*DeNardo et al., 2015*; *Kim et al., 2015*; *Kinnischtzke et al., 2016*; *Yamawaki et al., 2016*). However, input strength was consistently different across projection classes. When comparing looped IT neurons with neighboring subcortical-projecting PT or CT neurons, both FF and FB inputs always innervated looped IT neurons more strongly. Given the distinct gene-expression profiles of IT, PT, and CT neurons (*Tasic et al., 2018*), the preferential wiring of FF and FB axons to IT neurons could be guided by projection-type specific molecular cues. These findings accord with rabies virus (RV)-mediated trans-synaptic retrograde tracing experiments reporting that V1 IT neurons receive a larger proportion of their monosynaptic inputs from higher visual cortices than do V1 PT neurons (*Kim et al., 2015*). They are also consistent with studies showing that FF inputs from V1 to frontal cortices preferentially innervate looped neurons over PT neurons (*Zhang et al., 2016*) and that looped IT neurons in L6 of somatosensory cortex receive stronger input from motor cortex than subcortically projecting neurons (*Kinnischtzke et al., 2016*). Conversely, inputs from retrosplenial cortex to primary motor cortex do not show a preference for IT over CT or PT neurons (*Yamawaki et al., 2016*). Similarly, while we found a strong preference of V2L inputs for IT over CT neurons, single-cell RV tracings in V1 L6 found that CT neurons had a greater fraction of presynaptic partners from other cortical areas than IT neurons (*Vélez-Fort et al., 2014*). It remains to be elucidated to what extent these dissimilarities are due to differing selectivity across CC pathways for projection neuron types. Alternatively, since in many cases we found that projection-type specificity was present only for inputs terminating in the apical dendrites (*Figure 5*, *Figure 5—figure supplement 1*), selectivity for IT neurons may have been missed in previous studies using methods that cannot distinguish input strength in different dendritic compartments.

All the projections in this study contacted looped IT neurons, consistent with previous reports from the visual (*Johnson and Burkhalter, 1997*; *Kim et al., 2020*), somatosensory (*Kinnischtzke et al., 2016*), frontal (*Zhang et al., 2016*), and motor (*Mao et al., 2011*; *Yamawaki et al., 2016*) cortices. We extended these previous observations by comparing inputs to looped and non-looped IT neurons, and finding selectively stronger connections in looped over non-looped neurons in seven out of eight tested projections targeting infragranular layers. Thus, for all CC visual afferents in this study, IT neurons were the principal excitatory targets in deep layers, and in most of these cases, looped IT neurons were the main recipients.

The specificity of CC connections for looped neurons in infragranular layers was not absolute, as we detected monosynaptic CC inputs in all cell types. However, our measurements likely underestimate the selectivity of CC inputs for looped neurons. First, many, if not most, projection neurons

target multiple cortical areas (*Economo et al., 2016*; *Han et al., 2018*). Thus, since retrograde tracer injections only label a fraction of neurons projecting to the injection site, some non-looped neurons might in fact send a looped projection despite not being retrogradely labeled, thereby reducing our observed specificity for looped neurons. Second, looped connectivity would also be underestimated if it only involves CC afferents originating from a specific layer, as we expressed ChR2 non-specifically in projection neurons spanning multiple layers. Future experiments with layer-specific transgenic mouse lines (*Harris et al., 2019*) will make it possible to compare the specificity of projections with different laminar origins.

FF and FB connections to L2/3 neurons were consistently different from those targeting deeper neurons as they showed either no predilection for forming monosynaptic loops (three out of four projections), or were stronger, not weaker, when contacting non-looped IT neurons (V1→V2M). Pairs of L2/3 neurons were biased to upper L2/3 when measuring FB inputs and biased to lower L2/3 when measuring FF inputs (*Figure 2—figure supplement 1*). Thus, we cannot discard that FB connections also show a preference for non-looped neurons when targeting lower L2/3, as V1→V2M FF inputs do. While the axons of most L2/3 neurons branch to innervate multiple cortical areas, especially those of V1 neurons projecting to V2L (*Han et al., 2018*), the absence of loop specificity in L2/3 is unlikely to be caused by looped neurons being misattributed as non-looped neurons due to incomplete labeling from tracer injections. This is because L5 showed looped connectivity despite having a larger percentage of neurons with bifurcating axons (*Figure 1*). A recent report using RV-mediated monosynaptic input tracing found that looped L2/3 neurons in V1 had a greater percentage of presynaptic partner cells in visual areas to which they projected (*Kim et al., 2020*). Given that FB input strength in looped and non-looped L2/3 cells is similar, the presence of more presynaptic partners in the higher visual areas to which L2/3 neurons project would be consistent with our results if FB synapses in looped cells were fewer or weaker than in non-looped ones, since number of presynaptic partners does not necessarily reflect input strength. Thus, the connectivity rules of CC projections in L2/3 are different from those in deep layers, in which neurons consistently received stronger inputs from areas they innervated.

While, overall, the selectivity for looped neurons followed similar laminar-specific patterns across CC projections as described above, there was some variability in the connectivity rules (*Figure 7*). One possibility is that our measurements are underpowered to detect weaker selectivity in some projections that would result in a more coherent picture across cortical inputs. Alternatively, the selectivity of CC inputs for looped neurons might vary across individual cortical projections. This would be consistent with the stronger selectivity we observed for IT over PT/CT neurons in visual cortices and the lack of such selectivity reported in motor cortex (*Yamawaki et al., 2016*).

## A role for the apical tufts of IT neurons in looped hierarchical interactions

We found that both FF and FB inputs innervated the perisomatic and apical dendrites of L5 IT pyramidal neurons, but that they differed in their apical distribution. FB innervated the distal dendritic tufts in L1 while FF inputs extended along the apical shaft up to L2 (*Figure 3*, *Figure 3—figure supplement 1*). For both FF and FB projections to looped L5 neurons, we observed stronger selectivity for apical dendrites than for perisomatic ones (*Figure 5*, *Figure 7*). This preference was found in three out of the four CC projections studied (including in all FB pathways) when comparing looped with non-looped IT neurons, suggesting an important role for the apical dendritic trees of L5 IT neurons in many looped hierarchical interactions. Inputs to apical tufts might be selectively involved in recurrent computations, while perisomatic inputs might mediate other hierarchical exchanges. Through a dynamic interplay of active and passive membrane properties, the apical arbors of L5 pyramidal neurons can perform complex computations combining inputs from different dendritic compartments (*Larkum, 2013*; *London and Häusser, 2005*; *Stuart and Spruston, 2015*). The apical dendrites of L5 IT neurons are less well understood than those of PT neurons, being thinner and less experimentally tractable. Understanding hierarchical computations will require elucidating how looped L5 IT neurons integrate CC inputs in their distal tufts with inputs arriving in other dendritic regions.

While inputs to L2/3 neurons were for the most part equally strong in looped and non-looped IT neurons, FF inputs to V2M were selectively stronger, not weaker, when contacting dendrites of non-

looped IT neurons in L1 (*Figure 6D*). Thus, as with L5 neurons, distal input in L2/3 neurons might also be selectively involved in looped interactions, albeit with different rules.

## Functional implications

The stronger innervation of looped neurons resulted in the selective modulation of their spiking and subthreshold activity in an ex vivo functioning circuit (*Figure 8*), showing that the connectional selectivity measured using sCRACM has functional consequences in cortical networks. The selective engagement of looped neurons by both FF and FB afferents supports several theories advocating looped computations in CC circuitry (*Bastos et al., 2012*; *Guerguiev et al., 2017*; *Keller and Mrsic-Flogel, 2018*; *Mumford, 1992*; *Rao and Ballard, 1999*; *Richards and Lillicrap, 2019*; *Roelfsema and Holtmaat, 2018*; *Sacramento et al., 2018*). Our observations suggest that L5/6 IT neurons might be preferential players in the implementation of these long-range loops. CC inputs and postsynaptic neurons in their target area have similar tuning properties (*Glickfeld et al., 2013*; *Huh et al., 2018*; *Zhang et al., 2018*) and overlapping receptive fields (*Marques et al., 2018*). Thus, given the prominence of looped connectivity in FF and FB visual pathways, related visual signals are likely relayed back and forth between interconnected neurons located at different hierarchical levels.

According to recent models, looped FB innervation could allow neurons embedded in a hierarchical network to optimize synaptic weights toward the global desired output, akin to the backpropagation algorithm used to train artificial neural networks (*Guerguiev et al., 2017*; *Lillicrap et al., 2016*; *Richards and Lillicrap, 2019*; *Roelfsema and Holtmaat, 2018*; *Sacramento et al., 2018*). The apical dendrites of looped neurons are integral to many such models, wherein apical inputs trigger synaptic plasticity of basal inputs to instruct learning. Thus, selective targeting of the apical compartments of looped neurons by descending inputs, as observed here, may allow L5 IT neurons to update synaptic strengths based on activity in the higher-order areas that they project to. Such a role in looped interactions does not negate the involvement of apical dendrites in other non-looped computations. For example, we found that L5 PT neurons received weaker FB inputs in their distal tufts compared to L5 IT neurons (*Figure 5*), despite having more extensive dendrites in L1 (*Figure 5—figure supplement 2*). Since L5 PT neurons receive more inputs from frontal and associative areas than IT neurons (*Kim et al., 2015*), their thicker apical dendrites might be specialized in mediating top-down processes that do not require excitatory looped connectivity, such as brain state-dependent and attentional modulations of perceptual saliency (*Manita et al., 2015*; *Takahashi et al., 2020*; *Takahashi et al., 2016*; *Zhang et al., 2014*).

Models in which descending projections modulate plasticity in lower-order synapses to optimize the global output of the network result in alignment of FF and FB weights, such that synaptic strengths between pairs of reciprocally connected neurons tend to become similar (*Guerguiev et al., 2017*; *Lillicrap et al., 2016*; *Sacramento et al., 2018*). Whether the neocortex follows such rules is a matter of speculation (*Lillicrap et al., 2020*; *Richards and Lillicrap, 2019*; *Roelfsema and Holtmaat, 2018*; *Whittington and Bogacz, 2019*). In the presence of weight symmetry, FF and FB projection neurons would be expected to receive minimal input from areas they do not project to, and to receive strong input from areas that they do. Thus, alignment of synaptic weights between neuron pairs will lead to stronger average synaptic input to looped neurons from both ascending and descending pathways, as observed here. While our results are consistent with some degree of weight alignment in neocortical circuits, our experimental approach cannot prove the existence of symmetric synaptic weights in pairs of neurons at different hierarchical levels. It remains unknown whether the monosynaptic loops that we unveil here arise from pairs of neurons selectively targeting each other in a recurrent loop. Given that ChR2 was expressed in multiple presynaptic neurons, our experiments measured the selectivity of afferent populations and do not have the resolution to resolve interareal loops with single-cell resolution. There is evidence that L5/6 cortical neurons receive long-range CC inputs preferentially from L5/6 neurons (*DeNardo et al., 2015*). Moreover, L6 IT neurons innervate deep layers in their target areas (*Tasic et al., 2018*), and L5 IT neurons have access to corresponding L5 IT neurons in distant cortical regions, as indicated by their axonal arborization pattern (*Harris et al., 2019*; *Tasic et al., 2018*). Thus, while our findings and the laminar distribution of CC connections are consistent with pairs of same-layer neurons in different cortical areas selectively engaging in bidirectional monosynaptic loops, the prevalence of such a circuit arrangement has yet to be determined. Since input strength in L2/3 neurons was either independent of looped connectivity or weaker in looped neurons, synaptic weight symmetry, if present,

might selectively involve deep cortical layers and not superficial ones. Thus, hierarchical learning resulting in synaptic weight symmetry could also be laminar-specific.

Our observations reported here suggest that recurrent interareal cortical interactions may play different roles in supragranular and infragranular layers. They also provide a framework for future work on the role of projection neurons in different cortical layers in recurrent hierarchical processes.

# Materials and methods

## Key resources table

| Reagent type (species) or resource | Designation | Source or reference | Identifiers | Additional information |
|---|---|---|---|---|
| Strain, strain background (*Mus musculus*) | C57BL/6 | Jackson Laboratory | JAX:000664, RRID:IMSR_JAX:000664 | Bred in-house |
| Antibody | Anti-GFP (rabbit polyclonal) | Thermo Fisher | Catalog # A-6455, RRID:AB_221570 | (1:1000) |
| Antibody | Alexa Fluor 488-conjugated secondary antibody | Thermo Fisher | Catalog # A-11008, RRID:AB_143165 | (1:1000) |
| Recombinant DNA reagent | AAV-2/1-CAG-ChR2-Venus | Addgene | RRID:Addgene_20071 | |
| Recombinant DNA reagent | AAV5-CaMKIIa-hChR2(H134R)-EYFP | Addgene | RRID:Addgene_26969 | |
| Recombinant DNA reagent | AAV2/1-synapsin-EGFP | UPenn Vector Core | RRID:Addgene_105539 | |
| Peptide, recombinant protein | Cholera toxin B (Alexa Fluor 647) | Thermo Fisher | Catalog #: C34778 | 1 mg/ml |
| Chemical compound, drug | Red Retrobeads IX | Lumafluor | | |
| Software, algorithm | Ephus | Vidrio Technologies *Suter et al., 2010* | PMID:21960959 | |

## Animal surgeries

All procedures were reviewed by the Champalimaud Centre for the Unknown Ethics Committee and performed in accordance with the Portuguese Veterinary General Direction guidelines. Surgeries were conducted in either male or female C57BL/6J mice (P26–P28) under anesthesia (intraperitoneal, 37.5 mg/kg ketamine, 0.5 mg/kg medetomidine). Virus expressing ChR2 (AAV-2/1-CAG-Channelrhodopsin-2-Venus, Addgene #20071; 20–25 nl, titer ~$5\times10^{12}$ vg/ml) was delivered intracortically either to V1 to label FF projections or V2L/V2M to label FB projections, and co-injected with red-fluorescent microspheres (Red Retrobeads IX, Lumafluor; 10–12.5 nl) to retrogradely label cells projecting to the source of FF/FB input. A second retrograde tracer (Cholera toxin subunit B, Alexa Fluor 647 Conjugate, Thermo Fisher, 50–60 nl, 1.0 mg/ml) was injected elsewhere to label cells projecting to a different cortical or subcortical area. For axonal quantification, we used AAV2/1-synapsin-EGFP (Penn Vector Core p1696, ~$5\times10^{12}$ vg/ml). For *Figure 2—figure supplement 2*, we replaced AAV-2/1-CAG-Channelrhodopsin-2-Venus with AAV5-CaMKIIa-hChR2(H134R)-EYFP (Addgene #26969, ~$0.5\times10^{12}$ vg/ml) when injecting in V2M. Pulled glass injection pipettes (Drummond Scientific) tip diameter of 15–20 µm. Stereotaxic coordinates for V1 and V2L were measured from the midline and from the posterior-most point of the transverse sinus (lateral of midline/anterior of transverse sinus/depth in mm): V1 (2.3/1.3/0.775), V2L (3.5/1.7/0.9). Stereotaxic coordinates for V2M and SC were measured from the midline and from the sinus confluence, the point at which the transverse sinuses meet the superior sagittal sinus (lateral of midline/anterior of sinus confluence/depth in mm): V2M (1.6/1.25/0.8), SC (0.5/0.4/1.5 and 1.8). We verified the accuracy of V2L and V2M

coordinates by identifying the borders of visual areas relative to the injection sites in vivo in a subset of animals (*Figure 1—figure supplement 1*). Animals were injected in V2L and V2M with red-fluorescent latex microspheres as described above. Approximately 30 days later, a circular craniotomy was performed over the visual cortex (diameter: 4 mm) and an imaging window was embedded into the craniotomy and secured in place using black dental cement. A custom-designed iron headpost was attached to the skull with dental acrylic. Intrinsic signal imaging was performed to identify the position of the visual areas relative to the injection sites as previously described (*Garrett et al., 2014*; *Marques et al., 2018*; *Figure 1—figure supplement 1*). In 7/7 mice screened, both V2L and V2M injections were located outside the borders of V1. V2L injections targeted mainly the lateromedial (LM) visual cortex, while V2M injections labeled the anteromedial (AM) and/or anterior part of the posteromedial (PM) visual cortices (*Wang and Burkhalter, 2007*). To label CT neurons, injections of retrograde tracers were targeted to the dorsal lateral geniculate nucleus (dLGN) for V1 recordings, and the lateral posterior nucleus (LP) of the thalamus for V2L recordings. Stereotactic coordinates were measured from the midline and from bregma (lateral of midline/posterior of bregma/depth in mm): dLGN (2.3/1.75/2.8), LP (1.35/1.75/2.65 and 2.85). We cannot discard the possibility that cholera toxin injections were not entirely confined to either dLGN or LP and encompassed both nuclei in some cases. Animals were maintained at 37°C on a heating pad during surgery and returned to their home cages after surgery (maximum of five animals per cage). All animals were housed in a room with a regular 12 hr light/dark cycle.

## Slice preparation

Fourteen to 20 days (age range P40–P48) after the surgery, mice were decapitated under deep anesthesia (isoflurane) and brains were dissected in ice-cold choline chloride solution (110 mM choline chloride, 25 mM $NaHCO_3$, 25 mM D-glucose, 11.6 mM sodium ascorbate, 7 mM $MgCl_2$, 3.1 mM sodium pyruvate, 2.5 mM KCl, 1.25 mM $NaH_2PO_4$, and 0.5 mM $CaCl_2$ [Sigma]; aerated with 95% $O_2$/5% $CO_2$) and sectioned in 300-μm-thick coronal slices using a Leica VT1200S vibratome. Slices were then incubated for 30 min at 37°C in artificial cerebrospinal fluid (127 mM NaCl, 25 mM $NaHCO_3$, 25 mM D-glucose, 2.5 mM KCl, 1 mM $MgCl_2$, 2 mM $CaCl_2$, and 1.25 mM $NaH_2PO_4$ (Sigma); aerated with 95% $O_2$/5% $CO_2$).

## Electrophysiology and photostimulation

Neurons were patched with borosilicate pipettes (resistance 3–5 MΩ, Werner Instruments) filled with potassium gluconate intracellular solution (128 mM potassium gluconate, 4 mM $MgCl_2$, 10 mM HEPES, 1 mM EGTA, 4 mM $Na_2ATP$, 0.4 mM $Na_2GTP$, 10 mM sodium phosphocreatine, 3 mM sodium L-ascorbate, 3 mg/ml biocytin [Sigma] and 5 μg/ml Alexa Fluor 488 dye [Thermo Fisher Scientific]; pH 7.25, 290 mOsm). All sCRACM recordings were performed at room temperature (22–24°C) and with the presence of TTX (1 μM), CPP (5 μM), and 4-AP (100 μM) in the bath. For *Figure 5—figure supplement 4*, ZD7288 (10 μM) was also applied to control for $I_h$ differences between cell types. Areas V2L and V2M were identified by the presence of Venus-expressing FF axons. When measuring FB input strength, we verified that V2L or V2M injections resulted in the expected laminar distribution of Venus-expressing axons in V1 (*Figure 1C* and *Figure 2—figure supplement 2*) in the recorded slice. For sCRACM mapping, fluorescent-positive cells were recorded sequentially in voltage clamp (−70 mV) at depths of >30 μm in the same slice. Double-labeled cells were not recorded. A blue laser (473 nm, Cobolt Laser) was used for photostimulation to evoke EPSCs. Duration (1 or 4 ms) and intensity (0.1–1.1 μW) of light pulses were controlled with a Pockels cell (ConOptics) and a shutter (Thorlabs). The laser beam (diameter ~15 μm, not taking into account tissue scattering) was rapidly repositioned using galvanometer mirrors (Thorlabs) and delivered through an air immersion objective (Olympus 4X, NA 0.1) on either a 16 × 16 grid (L2/3 cells) or a 12 × 24 grid (L5 and L6 cells) with 50 μm spacing and 400 ms inter-stimulus interval. Stimuli were given in a spatial sequence pattern designed to maximize the time between neighboring locations. The stimulus pattern was flipped and rotated between maps to avoid sequence-specific responses. sCRACM maps were repeated two to five times for each cell. Laser power was manually adjusted in each experiment using a graduated neutral density filter (Edmund Optics) so that peak amplitudes smaller than 100 pA were evoked in the most excitable locations for the first recorded cell in the pair. Pairs of different projection neurons recorded at similar cortical depths in the same layer and in close proximity

(mean ± s.d.: 73.27 ± 47.41 µm) were photostimulated using the same laser power and pulse duration. The order in which cell types were recorded alternated between pairs. Data were acquired with a Multi clamp 700B amplifier (Axon Instruments) and digitalized with National Instruments acquisition boards controlled under Matlab using Ephus (*Suter et al., 2010*).

### Current-clamp experiments

To examine the functional impact of selective looped interactions (*Figure 8*), recordings were performed in current clamp in the absence of channel blockers and at near-physiological temperatures (34˚C). Neurons were photostimulated with 20 ms pulses from a blue LED (Cairn Research) through a water immersion objective (Olympus 60X, NA1.0, 20 µW at the focal point). Steps of positive current (0–200 pA, 500 ms duration) were injected every second. Trials with and without the LED pulses were interleaved. In trials with current injection, the LED light pulse began 200 ms after current step onset. We compared the mean amplitude of the LED-evoked excitatory postsynaptic potentials in trials without current injection to measure synaptic strength (*Figure 8C*). Spike frequency in a 0–50 ms window after LED onset was averaged across five trials and compared in *Figure 8E*.

### Immunohistochemistry and dendritic reconstructions

After whole-cell recordings, biocytin-filled neurons were fixed overnight in 4% paraformaldehyde (PFA) at 4˚C and transferred to phosphate-buffered saline the following day. Prior to staining, slices were rinsed in phosphate buffer (PB) 0.1 M. Endogenous peroxidases were quenched with 1% $H_2O_2$ (Sigma) in PB 0.1 M for 45 min at room temperature. Slices were rinsed again in PB 0.1 M and incubated in the ABC reaction (Vector Laboratories) for ~12 hr at room temperature (22–24˚C). After successive PB 0.1 M and Tris-buffered saline (TBS) washings, slices were subjected to the diaminobenzidine (DAB) reaction for 30–50 min (using 30 ml of TBS, 90 µl 3% $H_2O_2$, 225 µl of $NiCl_2$ (250 mM), and 7 mg of DAB [Sigma]). The DAB reaction was stopped with TBS. Slices were mounted and coverslipped with Mowiol mounting medium. Dendrites were reconstructed with Neurolucida software (MBF Bioscience) using the 40x magnification objective lens of an Olympus BX61 microscope. Tracings were imported into Matlab, corrected for shrinkage and analyzed using custom routines. Dendritic length density was calculated in 50 µm bins and interpolated for display.

### Laminar distribution of projection neuron subtypes and FF/FB axons

Injected animals were intracardially perfused with 4% PFA 14 days post-surgery and cryostat-sectioned in 20-µm-thick coronal slices. Slices were stained with 4′,6-diamidino-2-phenylindole dihydrochloride (DAPI) and imaged with the 20× objective of a Zeiss AxioImager M2 widefield fluorescence microscope. For quantification of different projection neurons in V1/V2L cortex (*Figure 1A, B*), three animals were used per dataset (eight slices per animal). Cells were counted within a 1000 × 1000 µm (V1) or 600 × 1000 µm (V2L) area. To normalize cell depth, fractional cortical depth (pia–cell distance/pia–white matter distance) was multiplied by average cortical thickness across the eight slices. For quantification of axons (*Figure 1C,D*), three animals were analyzed for both FF and FB datasets (eight slices per animal). Vertical fluorescence profiles of GFP-expressing axons were measured using ImageJ after subtracting background fluorescence from a hippocampal area devoid of labeled axons or somata.

### Quantification of retrograde infection of ChR2-expressing AAVs

We verified that AAV-2/1-CAG-Channelrhodopsin-2-Venus led to minimal retrograde infection in several ways (*Figure 2—figure supplement 2*). While Venus-labeled axons were clearly visible in target regions in unstained sections, Venus-positive somata were undetectable in these regions. In coronal slices stained with rabbit anti-GFP polyclonal antibody (1:1000 dilution, Thermo Fisher, catalog #A-6455) and Alexa Fluor 488-conjugated secondary antibody (1:1000 dilution, Thermo Fisher, catalog #A-11008) to boost Venus fluorescence, we detected a very small number of Venus-positive somata (*Figure 2—figure supplement 2*). In 20 µm sections of V1, we found 2.60 ± 0.94 Venus-expressing neurons/mm$^2$ (n = 3, V2M-injected mice). We also stained sections (n = 3 mice) for the neural marker NeuN (rabbit monoclonal, 1:1500 dilution, Abcam, catalog #ab177487), followed by an Alexa Fluor 647-conjugated secondary antibody (donkey polyclonal, 1:800 dilution, Jackson ImmunoResearch, catalog #711-605-152) and measured the density of neurons in V1 sections using

Cellpose (*Stringer et al., 2021*). From these counts we estimate that retrogradely infected neurons expressing ChR2 comprise 1 out of every 1282 ± 578 NeuN$^+$ V1 neurons (n = 3). In addition to these anatomical analyses, we also analyzed the onset of the sCRACM response in looped neurons to verify that they did not express ChR2 (*Figure 2—figure supplement 2*). We were able to detect early-onset (<2 ms) responses, consistent with a non-synaptic ChR2-induced current, in 10/235 (4.2%) of the recorded looped IT neurons. These 10 neurons were removed from further analysis. Together, these analyses suggest that, given their sparsity, collaterals of neurons retrogradely infected with ChR2 are unlikely to contribute to our measurements. We also conducted additional recordings using AAV5-ChR2 (AAV5-CaMKIIa-hChR2(H134R)-EYFP) to examine whether the small number of neurons retrogradely infected with AAV2/1-ChR2 could nevertheless be contributing to the stronger inputs in looped IT neurons (*Figure 2—figure supplement 2*). Consistent with previous results (*Kinnischtzke et al., 2014*), we found that the AAV5 resulted in less retrograde infection than AAV-2/1-CAG-Channelrhodopsin-2-Venus in sections immunostained with anti-GFP antibody (number of retrogradely infected neurons per mm$^2$ in 20-µm-thick V1 sections, 0.49 ± 0.20; fraction of NeuN$^+$ V1 neurons, 1 in 7009 ± 3434, n = 3 mice). None of the recorded looped IT neurons displayed early EPSC onset with AAV5 (n = 11, *Figure 2—figure supplement 2*). Despite the significantly lower density of retrogradely infected cells expressing ChR2 in AAV5 compared to AAV2/1 (AAV2/1-AAV5 density ratio = 5.6, p=0.0465, t-test), the relative strength of V2M→V1 inputs in looped vs. non-looped L6 IT neurons was similar using the two AAVs (*Figure 2—figure supplement 2*; total SRI, AAV2/1, −0.39 ± 0.36; AAV5, −0.52 ± 0.39, p=0.476). We conclude that local collaterals from the small number of neurons retrogradely infected with AAV2/1-ChR2 do not contribute significantly to the measurement of CC input strength.

## Data analysis

In electrophysiological recordings, the boundaries between layers were established as L1: pia−90 µm; L2/3: 90–350 µm; L4: 350–450 µm; L5: 450–650 µm; L6 650–950 µm. To correct for differences in cortical thickness due to variability in slicing angle of brain sections, the fractional cortical depth of each recorded neuron was positioned on a reference cortical slice with a thickness of 950 µm. Only pairs of cells with an intersomatic distance of <200 µm were included in analyses. In cases where multiple cells were recorded in the same slice, cells nearest each other were paired. Traces were baseline-substracted (baseline period: 50 ms before laser onset) and averaged across successive sCRACM maps. Average responses at each location were calculated as the mean EPSC 0–75 ms after laser onset and are therefore a measure of charge (*D'Souza et al., 2016*; *Petreanu et al., 2009*). In *Figure 3* and *Figure 3—figure supplement 1*, sCRACM maps were aligned by pia and soma position, respectively, and linearly interpolated for display. To calculate input strength, input at responsive locations (EPSC amplitude >5 standard deviations above baseline) was summed across the entire map or within the perisomatic or apical regions. The sign was flipped so that stronger inputs resulted in larger values. Pairs in which neither cell showed detectable input (total sCRACM input <20 pA) were discarded. To calculate the SRI, we required at least one cell with summed inputs >5 pA in the region of interest. The SRI was calculated as follows:

$$\frac{\sum Responses\ of\ non\ looped\ neuron\ -\ \sum responses\ of\ looped\ neuron}{\sum Responses\ of\ non\ looped\ neuron\ +\ \sum responses\ of\ looped\ neuron}$$

For inclusion in fraction-to-apical analyses, both cells in the pair required detectable input. Perisomatic and apical responses in neurons of different layers were calculated by summing inputs within a given cortical depth (L2/3: perisomatic 150–650 µm, L1: 0–50 µm; L5: perisomatic: 400–900 µm, L1: 0–150 µm for FB inputs, apical: 0–300 µm for FF inputs; L6: perisomatic: 600–950 µm, apical: 0–500 µm). For L5 datasets, pairs with any cut apical dendrites were discarded. Since many L2/3 neurons were located in upper L2/3, apical and perisomatic dendrites may be intermingled in L1 in some cases. Thus, responses in L1 could include inputs made on perisomatic and apical dendrites. We therefore refer to these as 'inputs in L1'. As the resolution of sCRACM is ~60 µm (*Petreanu et al., 2009*), to better distinguish inputs in L1 from those in L2/3, we removed responses in the two rows flanking the L1–L2/3 boundary when analyzing inputs in L1 and the perisomatic area of L2/3 neurons. For L6 neurons, we could not always verify the integrity of apical dendrites, thus when no difference in apical input is observed across cell types, this may be due to incomplete arbors. To compute

average traces in *Figures 4–6*, *Figure 4—figure supplement 1*, *Figure 5—figure supplement 1*, *Figure 5—figure supplement 4* and *Figure 6—figure supplement 1*, EPSCs at all locations in a given region of interest (perisomatic, apical, L1, or total) were averaged for each neuron, regardless of whether the location had a detectable light-evoked response, and subsequently averaged across neurons of the same projection class. To visualize the average subcellular distribution of inputs (*Figure 3*), only cells with sCRACM input >10 pA for at least one location were included.

To quantify relative synaptic strength in the current-clamp experiments (*Figure 8C*), we calculated the CRI as follows:

$$\frac{EPSP\ amplitude\ in\ non\ looped\ neuron - EPSP\ amplitude\ in\ looped\ neuron}{EPSP\ amplitude\ in\ non\ looped\ neuron + EPSP\ amplitude\ looped\ neuron}$$

SRI and CRI values were tested using the two-tailed Student's t-test, and fraction of inputs in the apical region were tested using the two-tailed Wilcoxon signed-rank test for paired samples. Data in the text is mean ± standard deviation. No statistical tests were used to predetermine the number of cell pairs, but our sample sizes are comparable to those in similar studies (*D'Souza et al., 2016*; *Kinnischtzke et al., 2016*; *Mao et al., 2011*; *Morgenstern et al., 2016*; *Yamawaki et al., 2016*; *Yang et al., 2013*). All statistical analyses were performed with Matlab.

### Simulations of passive dendritic filtering of L1 inputs

Traced dendrites of L5 neurons in V1 were imported into the NEURON simulation environment (*Hines and Carnevale, 1997*). We set the diameter of the somata, apical trunks, and apical tuft branches for the different projection neurons using manual measurements from biocytin-stained arbors (*Figure 5—figure supplement 2*). For each cell belonging to a given projection type, segments from the same dendritic compartment were assigned the same diameter value (apical trunk width, μm: SC-projecting, 1.7, V2M-projecting, 1.14; V2L-projecting, 1; apical tuft branch width, μm: SC-projecting, 0.7, V2M-projecting, 0.4, V2L-projecting, 0.43). The biophysical properties used for simulations were as follows: cytoplasmic resistivity, $R_a$ = 35.4 cm; specific membrane capacitance, $C_m$ = 1 F/cm$^2$; resting conductance, $g_{pas}$ = 1/20,000; resting potential, $E_{pas}$=–65 mV. We applied a synaptic density of 0.2 synapses per μm of apical dendritic segment when distributing passive synapses over the apical tree. The location of synapses along individual dendritic segments was randomly determined, and synaptic conductance was approximated by an alpha function, with parameters τ=0.1 ms and $g_{max}$ = 1 μS. For each neuron, we then simulated responses to apical tuft inputs under single-electrode voltage-clamp at the soma (−70 mV, 10 MΩ resistance), assuming no axonal selectivity for postsynaptic cell types. We conducted 100 simulations for each neuron and quantified the mean charge or amplitude measured at the soma (*Figure 5—figure supplement 3*). To measure input resistance (*Figure 5—figure supplement 3E*), we simulated somatic voltage-clamp recordings and measured input resistance using a −5 mV voltage step.

## Acknowledgements

We thank N Yamawaki, B Atallah, and M Fridman for critical comments on the manuscript and J Sacramento and S Keemink for discussions. This work was supported by fellowships from Fundação para a Ciência e a Tecnologia to HY and BB, and by grants from Marie Curie (PCIG12-GA-2012-334353), la Caixa Banking Foundation (LCF/PR/HR17/52150005), and Fundação para a Ciência e a Tecnologia (LISBOA-01-0145-FEDER 030328 and Congento LISBOA-01-0145-FEDER-022170), co-financed by FCT (Portugal) and Lisboa2020, under the PORTUGAL2020 agreement (European Regional Development Fund) and the Champalimaud Foundation.

## Additional information

### Funding

| Funder | Grant reference number | Author |
| --- | --- | --- |
| "la Caixa" Foundation | LCF/PR/HR17/52150005 | Leopoldo Petreanu |
| Fundação para a Ciência e a | LISBOA-01-0145-FEDER | Leopoldo Petreanu |

| | | |
|---|---|---|
| Tecnologia | 030328 | |
| Fundação para a Ciência e a Tecnologia | Congento LISBOA-01-0145-FEDER-022170 | Hedi Young<br>Beatriz Belbut<br>Margarida Baeta<br>Leopoldo Petreanu |
| Fundação para a Ciência e a Tecnologia | SFRH/BD/52221/2013 | Hedi Young |
| FP7 Marie-Curie Actions | PCIG12-GA-2012–334353 | Leopoldo Petreanu |
| Fundação para a Ciência e a Tecnologia | SFRH/BD/148468/2019 | Beatriz Belbut |

The funders had no role in study design, data collection and interpretation, or the decision to submit the work for publication.

### Author contributions

Hedi Young, Conceptualization, Data curation, Formal analysis, Investigation, Methodology, Writing - original draft, Writing - review and editing; Beatriz Belbut, Software, Investigation; Margarida Baeta, Data curation, Investigation, Visualization; Leopoldo Petreanu, Conceptualization, Resources, Formal analysis, Supervision, Funding acquisition, Visualization, Writing - original draft, Project administration, Writing - review and editing

### Author ORCIDs

Beatriz Belbut http://orcid.org/0000-0003-0341-0585
Leopoldo Petreanu https://orcid.org/0000-0003-1434-4691

### Ethics

Animal experimentation: All procedures were reviewed and performed in accordance with the Champalimaud Centre for the Unknown Ethics Committee and approved by the Portuguese Veterinary General Direction (Ref.No.0421/000/000/2019).

### Decision letter and Author response

Decision letter https://doi.org/10.7554/eLife.59551.sa1
Author response https://doi.org/10.7554/eLife.59551.sa2

## Additional files

### Supplementary files

• Transparent reporting form

### Data availability

All data are publicly available on Dryad https://doi.org/10.5061/dryad.1ns1rn8r7.

The following dataset was generated:

| Author(s) | Year | Dataset title | Dataset URL | Database and Identifier |
|---|---|---|---|---|
| Young H, Belbut B, Baeta M, Petreanu L | 2020 | Laminar-specific cortico-cortical loops in mouse visual cortex | https://doi.org/10.5061/dryad.1ns1rn8r7 | Dryad Digital Repository, 10.5061/dryad.1ns1rn8r7 |

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
