## [Decision Letter]

**Acceptance summary:**

This article determines the synaptic organization of feed-forward and feed-back connections between higher order and primary visual cortex. The authors find that feed-forward and feed-back connections generally form stronger connections onto cortical neurons in layers 5/6 that project back to the source as compared to neurons that project to other cortical or subcortical targets. In contrast, this specificity was not observed for layer 2/3 neurons, suggesting distinct organizing principles for supragranular and infragranular layers.

**Decision letter after peer review:**

Thank you for submitting your article "Laminar-specific cortico-cortical loops in mouse visual cortex" for consideration by *eLife*. Your article has been reviewed by three peer reviewers, and the evaluation has been overseen by a Reviewing Editor and John Huguenard as the Senior Editor. The reviewers have opted to remain anonymous.

The reviewers have discussed the reviews with one another and the Reviewing Editor has drafted this decision to help you prepare a revised submission.

As the editors have judged that your manuscript is of interest but, as described below, that additional experiments are required before it is published, we would like to draw your attention to changes in our revision policy that we have made in response to COVID-19 (https://elifesciences.org/articles/57162). First, because many researchers have temporarily lost access to the labs, we will give authors as much time as they need to submit revised manuscripts. We are also offering, if you choose, to post the manuscript to bioRxiv (if it is not already there) along with this decision letter and a formal designation that the manuscript is "in revision at *eLife*". Please let us know if you would like to pursue this option. (If your work is more suitable for medRxiv, you will need to post the preprint yourself, as the mechanisms for us to do so are still in development.)

Summary:

This article investigates feed-forward (FF) and feed-back (FB) connections between higher-order (LM and AM) and lower-order (V1) visual cortex in the mouse. Experiments combined subcellular Channelrhodopsin-2-assisted circuit mapping (sCRACM) with retrograde tracers for identifying different types of cortical projection neurons to compare the connectivity of long-range corticocortical input onto the somas and dendrites of different cell types in layers 2/3, 5 and 6 (L2/3, L5, L6) including intratelencephalic (IT), pyramidal tract (PT) and corticothalamic (CT) neurons. The authors find that both FF and FB corticocortical connections make stronger excitatory connections onto L5/6 neurons projecting back to the source ("looped" IT neurons) than onto adjacent cells projecting to other targets ("non-looped" IT, PT or CT cells). In contrast, they did not observe equivalent specificity for "looped" over "non-looped" neurons in L2/3. Furthermore, for L5 neurons, they find some differences in the distribution of FF and FB input onto the apical oblique dendrites and distal apical dendrites of "looped" IT cells.

Essential revisions:

1) The reviewers agreed that central to any revision would be experiments addressing the functional consequences of these circuits. Performing current-clamp experiments in the absence of 4AP and TTX, showing not just preferential targeting but also activation of "looped" IT cells, performed at near-physiological temperatures, would help address this concern. Demonstrating a consequence of the dendritic targeting, perhaps in generating dendritic spikes, would also address the functional consequences of these circuits.

2) It is somewhat confusing to include CT and PT neurons in "non-looped" cells as the presentation implies that the lack of input is *because they are non-looped*, but it could be because they are CTs and PTs (just as interneurons or glia would be non-looped). A related point is that, in both the text and the figure panels, it would be much better to start with the non-looped IT experiments (rather than CT or PT), since those are the most important and interesting comparison with looping CC circuits. Experiments comparing CC input to looping/non-looping ITs is more of an apples-to-apples comparison and is more informative for the proposed circuit organization that for the looping-IT versus CT or PT comparison. Although these points are somewhat clarified in the Discussion, addressing these points in the Results and figure organization would improve the manuscript.

3) There was also concern regarding the use of AAV2/1 ChR2 for these experiments. The analyses assume that long-range connections are being selectively activated. However, AAV2/1 can generate some retrograde labeling (i.e. Kinnischtzke et al., 2014), raising the possibility that local connections could be contributing to the findings. The authors should demonstrate there is no retrograde labeling, which would be a major confound to their experiments. Anatomical analyses alone may not be enough sufficient to fully address this point; demonstrating that retrogradely labeled cells have no photocurrent should be sufficient.

4) In Figures 2 and 5, the L2/3 cells for testing FF input are biased towards L3, whereas those for FB analysis are biased towards L2. Similarly, the distribution of the neurons in L6 are also skewed with L6 IT biased towards the top and L6 CT biased to the bottom. These biases make it difficult to compare the populations, particularly since the circuit organization is different in L2 and L3 as well as in upper L6 and lower L6. Sampling more broadly across the two layers would address these biases.

5) The authors should add comparisons of the dendritic and somatic input to L2/3 and L6 neurons to compare with the findings regarding the somatic and dendritic inputs to L5 cells. While the authors say there is more FB input to dendrites (Figure 2 and subsection “FF and FB inputs innervate specific dendritic compartments of projection neurons”), this is not explored in Figure 5. Related, would it be possible in the subsection “FF and FB inputs are selectively stronger in looped L5 neurons” to add a statement about the overall results without the separation into apical/perisomatic?

6) As the experiments rely on the accuracy of the injections for ChR2-expression and retrograde labeling, the authors should provide additional images demonstrating their virus and retrobead injections and state in the Materials and methods whether and how the accuracy of their injections was confirmed after slicing. The authors should also include data on how the accuracy of LM and AM coordinates was verified by intrinsic signal imaging in a subset of animals, as briefly stated in the Materials and methods. This is of particular concern since LM and AM are relatively small structures closely adjacent to V1 (Wang and Burkhalter, 2007), thus off-target injections or spillover into V1 could potentially result in V1 cells expressing ChR2. Similarly, bead injections targeting LGN and LP, which are closely adjacent to one another, are also challenging.

7) Additionally, differences between the findings reported here and previous anatomical work in the same system raise concerns about the ability to detect all functional synapses using the sCRACM method (i.e. Velez-Fort et al., 2014; Velez-Fort et al., Neuron, 2018). Performing these experiments in current-clamp, in the absence of TTX and 4AP, and at near-physiologic temperatures, would address concerns that some inputs are not detectable using sCRACM.

8) The authors should state the number of animals and the number of neurons/pairs recorded in each associated figure legend. The authors should avoid saying "significant" unless they make an explicit comparison and provide relevant statistics (i.e. in the subsection “FF and FB inputs innervate specific dendritic compartments of projection neurons”) or describing differences as being present or "trending" when they do not reach statistical significance. Is it possible to put error estimates on the ratios? Calculating the median of the ratios is perhaps appropriate where differences are large, but for more subtle differences would the findings still hold if other measures are used, e.g. geometric means? There are very many connections being tested in this study with noise in the datasets, so if some comparisons do not "reach" significance, it does not necessarily undermine the main conclusions; in fact, it may point to some interesting differences between input types.

---

## [Author Response]

Essential revisions:1) The reviewers agreed that central to any revision would be experiments addressing the functional consequences of these circuits. Performing current-clamp experiments in the absence of 4AP and TTX, showing not just preferential targeting but also activation of "looped" IT cells, performed at near-physiological temperatures, would help address this concern. Demonstrating a consequence of the dendritic targeting, perhaps in generating dendritic spikes, would also address the functional consequences of these circuits.

We have performed new current-clamp experiments in the absence of channel blockers and at near-physiological temperatures. These show that stronger synaptic inputs, as measured under sCRACM conditions, do have functional consequences.

We found that photoactivation of FB inputs in a functioning circuit results in selective modulation of subthreshold and spiking activity in looped neurons in L6 (Figure 8).

2) It is somewhat confusing to include CT and PT neurons in "non-looped" cells as the presentation implies that the lack of input is because they are non-looped, but it could be because they are CTs and PTs (just as interneurons or glia would be non-looped). A related point is that, in both the text and the figure panels, it would be much better to start with the non-looped IT experiments (rather than CT or PT), since those are the most important and interesting comparison with looping CC circuits. Experiments comparing CC input to looping/non-looping ITs is more of an apples-to-apples comparison and is more informative for the proposed circuit organization that for the looping-IT versus CT or PT comparison. Although these points are somewhat clarified in the Discussion, addressing these points in the Results and figure organization would improve the manuscript.

We have reorganized the manuscript and figures as suggested. We now discuss and display the looped vs. non-looped IT comparison before the looped IT vs. PT/CT comparison. We also now use the term “non-looped” exclusively for IT cells to make the distinction between IT and CT/PT neurons clearer.

3) There was also concern regarding the use of AAV2/1 ChR2 for these experiments. The analyses assume that long-range connections are being selectively activated. However, AAV2/1 can generate some retrograde labeling (i.e. Kinnischtzke et al., 2014), raising the possibility that local connections could be contributing to the findings. The authors should demonstrate there is no retrograde labeling, which would be a major confound to their experiments. Anatomical analyses alone may not be enough sufficient to fully address this point; demonstrating that retrogradely labeled cells have no photocurrent should be sufficient.

We addressed the concerns regarding the possible contribution of local connections in our measurements in several ways. Using NeuN and anti-GFP staining in slices, we quantified the fraction of retrogradely-infected neurons expressing ChR2. We found that when using AAV2/1, only 1/1282 V1 neurons express ChR2 after virus injections in V2M.

As suggested, we also analyzed early-onset photocurrents in looped neurons retrogradely labeled with red beads. We found that only a small fraction (4.2%, 10/235) of looped cells recorded had early-onset responses (<2 ms from laser onset) using AAV2/1, confirming that the number of retrogradely-labeled neurons expressing ChR2 is very low. We removed these 10 cells from the dataset. To address whether the local collaterals of retrogradely-transfected cells could nevertheless be contaminating our recordings of non-transfected cells, despite their low number, we performed an additional experiment. We repeated our measurements of relative synaptic input strength in V2MàV1 inputs to looped and non-looped IT neurons in L6 with an AAV5-CamKII-h1ChR2(H134R)-EYFP virus. Using this virus, we found no cells with photocurrent and GFP antibody-stained histological sections contained even fewer numbers of retrogradely-infected neurons expressing ChR2 compared to AAV2/1 (1/7009 V1 neurons, p=0.0465), consistent with previous reports (Kinnischtzke, 2014). In the AAV5 sCRACM experiment, looped L6 IT cells were again innervated more strongly by V2M FB axons, and the effect was similar in magnitude to AAV2/1 (Figure 2—figure supplement 2, compare with Figure 4—figure supplement 1H). Together, these new experiments and analyses make it unlikely that local inputs are affecting our results. We now discuss these controls in the text and in a new section in the Materials and methods.

4) In Figures 2 and 5, the L2/3 cells for testing FF input are biased towards L3, whereas those for FB analysis are biased towards L2. Similarly, the distribution of the neurons in L6 are also skewed with L6 IT biased towards the top and L6 CT biased to the bottom. These biases make it difficult to compare the populations, particularly since the circuit organization is different in L2 and L3 as well as in upper L6 and lower L6. Sampling more broadly across the two layers would address these biases.

The biases in the laminar distribution of the recorded pairs in FF and FB experiments reflect the different distribution of the projection neurons themselves. In order to compare input strength, recorded pairs were required to be <200 µm from each other. While the distribution of IT neurons in L6 is biased toward upper L6, CT neurons have an opposite bias toward lower L6 (Figure 1). As a consequence, the depth of the recorded IT-CT pairs is biased toward the middle of the layer (Figure 2—figure supplement 1), where the two distributions are more likely to overlap. The same applies to IT-IT pairs recorded in L2/3. While FB-projecting L2/3 neurons in V2L are more broadly distributed in the layer, FF-projecting L2/3 neurons in V1 are mostly located in upper L2/3 (Figure 1). Due to these anatomical biases, it is very challenging to record a large enough number of paired neurons whose somata are <200 µm apart (the distance required to make a paired comparison) without having depth biases across the different projections and areas. However, now when we compare FF and FB inputs to L2/3 in the Discussion, we note that differences could be due to cortical depth biases in the populations.

5) The authors should add comparisons of the dendritic and somatic input to L2/3 and L6 neurons to compare with the findings regarding the somatic and dendritic inputs to L5 cells. While the authors say there is more FB input to dendrites (Figure 2 and subsection “FF and FB inputs innervate specific dendritic compartments of projection neurons”), this is not explored in Figure 5. Related, would it be possible in the subsection “FF and FB inputs are selectively stronger in looped L5 neurons” to add a statement about the overall results without the separation into apical/perisomatic?

We have now performed comparisons of total input, in addition to comparisons of apical and perisomatic input, when discussing data from all 3 layers. We have also changed the main figures so that similar analyses are made across layers and cell types (apical/perisomatic comparisons in the main figures, total input comparisons in the supplementary figures).

6) As the experiments rely on the accuracy of the injections for ChR2-expression and retrograde labeling, the authors should provide additional images demonstrating their virus and retrobead injections and state in the Materials and methods whether and how the accuracy of their injections was confirmed after slicing. The authors should also include data on how the accuracy of LM and AM coordinates was verified by intrinsic signal imaging in a subset of animals, as briefly stated in the Materials and methods. This is of particular concern since LM and AM are relatively small structures closely adjacent to V1 (Wang and Burkhalter, 2007), thus off-target injections or spillover into V1 could potentially result in V1 cells expressing ChR2. Similarly, bead injections targeting LGN and LP, which are closely adjacent to one another, are also challenging.

We have added a new supplementary figure with additional images of the injections. We include an example intrinsic image, with visual area borders overlaid, as well as example coronal sections showing the injection sites (Figure 1—figure supplement 1). While intrinsic imaging in all the cases we analyzed showed that injections were always outside V1, we did find AM injections that likely spread to the anterior part of PM (Figure 1—figure supplement 1). Similarly, while LM injections were in all cases focused in LM, we cannot discard some spill-over into AL. We now refer to injection sites as V2L and V2M to better reflect the possibility that they may encompass regions outside LM and AM, respectively. We also clarify in the Materials and methods section that V2L injections might include both LM and AL and that V2M injections might label both AM and PM.

Similarly, we agree that thalamic injections might include both dLGN and LP, so we now state that retrogradely-labeled CT neurons project to the visual thalamus, and not to a specific thalamic nucleus. We also clarify this in the Materials and methods.

We also state in the Materials and methods that the accuracy of the V2L and V2M injections was confirmed after slicing by the presence of typical FB laminar termination pattern of labeled axons in the acute brain slices. We now show the pattern of V2MàV1 inputs in Figure 2—figure supplement 2, in addition to that of V2LàV1 inputs in Figure 1.

7) Additionally, differences between the findings reported here and previous anatomical work in the same system raise concerns about the ability to detect all functional synapses using the sCRACM method (i.e. Velez-Fort et al., 2014; Velez-Fort et al., Neuron, 2018). Performing these experiments in current-clamp, in the absence of TTX and 4AP, and at near-physiologic temperatures, would address concerns that some inputs are not detectable using sCRACM.

We have now performed current-clamp experiments at near-physiological temperatures. The sign and magnitude of V2M FB inputs to looped IT neurons in L6 was similar to that measured under sCRACM conditions, showing that input strength using sCRACM accurately reflects synaptic strength under more naturalistic conditions (Figure 8). We now discuss this in the text.

8) The authors should state the number of animals and the number of neurons/pairs recorded in each associated figure legend.

We have added the number of animals and the number of neuron pairs to the figure legends/panels.

The authors should avoid saying "significant" unless they make an explicit comparison and provide relevant statistics (i.e. in the subsection “FF and FB inputs innervate specific dendritic compartments of projection neurons”) or describing differences as being present or "trending" when they do not reach statistical significance.

We have corrected the wording as suggested.

Is it possible to put error estimates on the ratios? Calculating the median of the ratios is perhaps appropriate where differences are large, but for more subtle differences would the findings still hold if other measures are used, e.g. geometric means?∑responsesofNonLoopedNeuron−∑responses of Looped Neuron∑responsesofNonLoopedNeuron+∑responses of Looped NeuronThere are very many connections being tested in this study with noise in the datasets, so if some comparisons do not "reach" significance, it does not necessarily undermine the main conclusions; in fact, it may point to some interesting differences between input types.

We now also discuss more explicitly the differences in the connectivity across projections of the same type.